# MergOPT: A Merge-Aware Optimizer for Robust Model Merging

**Enneng Yang**[1,2*]**, Qun Yang**[3*]**, Peng Wang**[1]**, Anke Tang**[4]**, Guibing Guo**[5]**, Xiaochun Cao**[1]**, Li Shen**[1,2†]

[1] Shenzhen Campus of Sun Yat-sen University
[2] Guangdong Laboratory of Artificial Intelligence and Digital Economy (SZ)
[3] National University of Defense Technology; [4] Wuhan University; [5] Northeastern University, China
ennengyang@gmail.com, yangqun@nudt.edu.cn, mathshenli@gmail.com

## Abstract

Model merging aims to integrate multiple independently fine-tuned expert models into a single model while preserving the knowledge of all experts. However, existing approaches mainly address parameter conflicts at the merging stage and overlook the role of the fine-tuning process, which often leads to significant post-merge performance degradation. To address this limitation, we propose a novel **merging-aware optimizer** (abbreviated as `MergOPT`) that injects principled merge-induced parameter shifts into the weight update steps so that the fine-tuned model exhibits a more stable loss landscape under subsequent merging operations. Specifically, we first formulate model merging as a distributionally robust optimization problem in the weight space: the parameters of other experts to be merged are viewed as adversarial merge-offsets, and fine-tuning adapts to the worst-case merging scenario. Building on this formulation, we analyze the distribution of parameter updates and the effects of merging hyperparameters, from which we derive a merging-guided feasible region for weight shifts. Finally, extensive experiments across four large language models (LLMs) and one vision model show that our approach consistently outperforms standard fine-tuning, yielding an average relative gain of 3.5% and a maximum gain of 9.5% across four merging strategies when merging seven experts.

## 1 Introduction

Multi-task learning is the conventional approach for adapting a foundation model to multiple downstream tasks, where diverse datasets are jointly used to update the model (Zhang & Yang, 2021; Chen et al., 2024). However, this strategy requires centralized access to data, leading to high management costs and privacy concerns. To overcome these limitations, model merging has recently been proposed as an alternative paradigm (Yang et al., 2024a). In this setting, multiple expert models are first fine-tuned independently on different tasks and then merged into a single model at the parameter level, with the goal of inheriting the knowledge of all experts without centralized data sharing. This paradigm has shown promising results in various domains, including computer vision (Ilharco et al., 2023; Ortiz-Jimenez et al., 2023; Jin et al., 2024; Gargiulo et al., 2025) and natural language processing (Yadav et al., 2023; Wan et al., 2024a; Yu et al., 2024; Akiba et al., 2025). A central challenge in model merging lies in effectively mitigating parameter conflicts that arise when integrating multiple expert models, as such conflicts often lead to severe performance degradation.

The most straightforward merging strategy is linear interpolation, where model parameters are simply averaged across experts (Wortsman et al., 2022). However, due to the highly nonlinear nature of deep neural networks and the complex interdependencies between tasks, this approach typically yields suboptimal results. To overcome these limitations, recent research has explored more sophisticated merging strategies. For instance, task arithmetic (Ilharco et al., 2023) first represents the difference between a fine-tuned model and its pre-trained counterpart as a task vector, and constructs a unified multi-task model by linearly combining these vectors, thereby partially preserving task-specific knowledge. In addition, adaptive weighting methods dynamically adjust the contribution of each

---

[*]Equal contribution
[†]Corresponding author

task according to task characteristics, employing either heuristic approaches such as evolutionary search (Akiba et al., 2025; Mencattini et al., 2025) or data-driven weight optimization (Matena & Raffel, 2022; Jin et al., 2023; Yang et al., 2024b; Tang et al., 2024a). Another line of work, subspace merging methods, alleviates task interference by projecting parameters into a sparse (Yadav et al., 2023; Yu et al., 2024; Wang et al., 2024; Zhu et al., 2024) or low-rank subspace (Gargiulo et al., 2025; Marczak et al., 2025), thereby mitigating performance loss caused by conflicts. Lastly, weight alignment methods exploit the property of linear mode connectivity in deep neural networks, which suggests that multiple equivalent loss landscapes can exist (Garipov et al., 2018; Draxler et al., 2018). By parameter permutations or aligning the parameters of expert models such that they lie within the same loss landscape, these approaches aim to reduce potential conflicts during model merging (Jordan et al., 2023; Ainsworth et al., 2023; Rinaldi et al., 2025). By leveraging fine-grained parameter manipulation, these methods typically achieve better performance than simple averaging.

Even with these advances, most of the existing methods predominantly focus on *the merging stage*, i.e., designing strategies to reduce conflicts when merging expert models, while largely overlooking the role of *the fine-tuning stage*. In this work, we contend that the effectiveness of the final merged model crucially depends on both stages: fine-tuning must prepare models in a way that facilitates compatibility, while merging must integrate them effectively. To the best of our knowledge, only a few model merging works explicitly focus on the fine-tuning stage. For example, tangent-space fine-tuning methods linearize the model and perform optimization in its tangent space to enhance weight disentanglement (Ortiz-Jimenez et al., 2023; Jin et al., 2024; Tang et al., 2024b), thereby alleviating conflicts during merging. However, inference with such linearized models typically incurs a $2-3\times$ higher computational cost compared to standard models (Ortiz-Jimenez et al., 2023). In another line of work, SAFT-Merge (Lee et al., 2025) is inspired by sharpness-aware minimization (SAM) (Foret et al., 2021; Kwon et al., 2021) and aims to improve mergeability by encouraging flatter loss landscapes during fine-tuning. Yet SAM-based fine-tuning usually doubles the training time relative to standard fine-tuning. Given that most existing approaches focus primarily on the merging stage and that the few methods targeting fine-tuning often come with substantial training or inference overhead, we argue that there is a strong need for a fine-tuning scheme that is both efficient and effective, while further improving the overall performance of model merging.

To address this limitation, this paper proposes a novel optimization approach for the fine-tuning stage, called **Merg**ing-Aware **Opt**imizer (referred to as `MergOPT`), specifically designed to produce expert models that are more amenable to merging. More specifically, the core idea of `MergOPT` is to formalize the parameter merging process as a merge-induced parameter offset operation and explicitly construct it during training as a distributionally robust optimization (DRO) (Lin et al., 2022) problem in weight space. In other words, the parameters (or task vectors) from other expert models to be merged can be regarded as merge-induced parameter offsets applied to a target model's parameters. The training objective is then to optimize against the worst-case merge-induced offset within the feasible region of this space, thereby improving the stability and effectiveness of the model during merging. To this end, we further specify the feasible region over these merge-offset configurations. Specifically, we decompose it into three dimensions: the distribution of task vectors, the range of merging coefficients, and the number of models to be merged. However, when training a single expert model, these three components are usually unknown. To solve this problem, we conduct an empirical analysis of task vectors. Results across three mainstream LLM architectures and seven real-world tasks demonstrate that each task vector can be well approximated by a Laplacian distribution (Kotz et al., 2012). For merging coefficients and model numbers, we define discrete feasible regions grounded in empirical observations and prior experience, ensuring both the practicality and interpretability of the resulting merge-offset space. Finally, we evaluate `MergOPT` on four LLM architectures of different scales (Llama 1B & 3B & 8B, Qwen 1.5B) combined with four representative model merging methods, applied to multiple downstream expert models. Experimental results demonstrate that `MergOPT` delivers substantial performance gains, with average relative improvements of about 3.5% and up to 9.5% when merging seven experts across four strategies, thereby validating its effectiveness in enhancing the robustness and practicality of model merging.

The **main contributions** of this work are summarized as follows: ❶ We highlight a critical yet underexplored aspect of model merging: the fine-tuning stage. We argue that the effectiveness of the merged model depends on both fine-tuning and merging, and emphasize the need for a dedicated fine-tuning scheme to improve model compatibility. ❷ We propose `MergOPT`, a merging-aware optimizer that formalizes merging as a merge-induced parameter offset in weight space

and applies distributionally robust optimization to enhance stability and effectiveness. We further define the feasible region of these merge-offset configurations through analysis of task vectors, merging coefficients, and model numbers. ❸ We perform extensive experiments on four large-scale LLM architectures and one vision model with four representative merging methods across seven downstream tasks. Results show that `MergOPT` consistently outperforms standard fine-tuning, demonstrating its effectiveness in improving the robustness and utility of model merging.

## 2 RELATED WORK

**Methods in the Merging Phase.** The most straightforward strategies are weight averaging (Wortsman et al., 2022) or task arithmetic (Ilharco et al., 2023), but their performance is often limited due to potential conflicts among models. To address this issue, more advanced merging techniques have been developed, which can be broadly categorized into three families: weighted merging, subspace-based merging, and weight alignment merging. (i) importance-based weighting methods aim to balance the contributions of different models using strategies such as grid search (Ilharco et al., 2023; Yadav et al., 2023), evolutionary algorithms (Akiba et al., 2025), or data-driven adaptive weighting (Matena & Raffel, 2022; Jin et al., 2023; Yang et al., 2024b; Tang et al., 2024a). For example, Fisher merging (Matena & Raffel, 2022) leverages Fisher information to assign parameter importance, while AdaMerging (Yang et al., 2024b) optimizes merging coefficients with unlabeled test data. (ii) subspace-based methods mitigate conflicts and reduce computational overhead by discarding redundant information and constraining merging to low-rank or sparse subspaces. Such as TIES-Merging (Yadav et al., 2023) and DARE (Yu et al., 2024), remove a large portion of unimportant parameter updates and adjust the remaining ones through sign aligning or rescaling. (iii) Weight alignment methods apply parameter permutations to obtain functionally equivalent solutions that lie in different loss basins (Jordan et al., 2023; Ainsworth et al., 2023; Rinaldi et al., 2025). By adjusting expert models to lie within the same basin, merging can typically be performed more smoothly and effectively.

**Methods in the Fine-Tuning Phase.** However, while these methods introduce clever designs at the merging stage, they still rely on standard optimizers and largely overlook the importance of preparing models to facilitate subsequent merging. To the best of our knowledge, only a few works focus on how to train models that are more amenable to merging. Ortiz-Jimenez et al. (2023) first pointed out that weight disentanglement is a key factor for the effectiveness of task-arithmetic-based model merging. Their method amplifies weight disentanglement by linearizing the model and performing fine-tuning in its tangent space (Jin et al., 2024; Tang et al., 2024b; Liu et al., 2024). Nevertheless, inference with the linearized model typically takes about two to three times longer than with its original nonlinear counterpart (Ortiz-Jimenez et al., 2023). In addition, our experimental results show that such approaches are still inferior in performance to the method proposed in this paper. SAFT-Merge (Lee et al., 2025) enhances mergeability during fine-tuning by applying sharpness-aware minimization. However, its training cost is nearly twice that of standard fine-tuning, making it inefficient for large models or datasets. In contrast, our `MergOPT` matches the efficiency of a standard optimizer while explicitly simulating merging via cross-expert merge-offsets, thereby improving stability and overall merging performance. It is worth mentioning that while most existing methods have been evaluated on vision models and image classification tasks, our work is conducted in the context of LLMs and text generation tasks.

## 3 METHOD

In this section, we first introduce the preliminaries and notations used in this paper (Sec. 3.1). Then, we present our proposed method, merge-aware fine-tuning via weight-space robust optimization, which aims to enhance the robustness of model merging (Sec. 3.2).

### 3.1 PRELIMINARIES

**Fine-Tuning from a Pre-Trained Model.** Let $\theta_0 \in \mathbb{R}^d$ denote the parameters of a pre-trained base model. We denote the training dataset as $\mathcal{D}_k^{\text{train}} = \{(x_i, y_i)\}_{i=1}^N$, where $x_i$ is the input and $y_i$ is the corresponding label for the $i$-th sample in task $k$ ($k \in \{1, 2, \ldots, K\}$). The fine-tuned parameters are denoted as $\theta_k \in \mathbb{R}^d$. The loss function is represented as $\ell_k(\theta_k; (x, y))$, which measures the

discrepancy between the model's prediction and the true label for a given input. The expected empirical risk on the training dataset is defined as:

$$\mathcal{L}_{\text{task}}(\theta_k; \mathcal{D}_k^{\text{train}}) = \mathbb{E}_{(x,y) \sim \mathcal{D}_k^{\text{train}}} \left[ \ell_k(\theta_k; (x,y)) \right]. \tag{1}$$

**Model Merging.** Model merging aims to combine multiple fine-tuned models into a single merged model $\theta_{\text{merged}}$. Given a set of $K$ fine-tuned models with parameters $\{\theta_k\}_{k=1}^K$, a common approach is to use task arithmetic (Ilharco et al., 2023):

$$\theta_{\text{merged}} = \theta_0 + \alpha \sum_{k=1}^K \Delta\theta_k, \tag{2}$$

where $\theta_0$ is the base model, $\Delta\theta_k = \theta_k - \theta_0$ denotes the task vector for task $k$, and $\alpha > 0$ is a scaling factor. The objective of the merged model is to achieve consistently low test loss across all tasks. Formally, the expected test risk of the merged model is defined as:

$$\mathcal{L}_{\text{merge}}(\theta_{\text{merged}}; \{\mathcal{D}_k^{\text{test}}\}_{k=1}^K) = \frac{1}{K} \sum_{k=1}^K \mathbb{E}_{(x,y) \sim \mathcal{D}_k^{\text{test}}} \left[ \ell_k(\theta_{\text{merged}}; (x,y)) \right], \tag{3}$$

where $\mathcal{D}_k^{\text{test}}$ is the test dataset for task $k$.

**Distributionally Robust Optimization (DRO).** DRO seeks to optimize model parameters under distributional uncertainty by minimizing the worst-case expected loss over a family of probability distributions $\mathcal{P}$ that are close to the empirical data distribution (Lin et al., 2022). Formally, the DRO objective can be expressed as:

$$\min_{\theta_k} \sup_{P \in \mathcal{P}} \mathbb{E}_{(x,y) \sim P} \left[ \ell_k(\theta_k; (x,y)) \right]. \tag{4}$$

Here, $\mathcal{P}$ denotes an ambiguity set of candidate distributions around the empirical distribution. $\mathcal{P}$ is typically defined by imposing constraints based on a divergence metric (e.g., Wasserstein distance, KL divergence), which controls the proximity between $P$ and the empirical distribution. In this work, unlike conventional DRO that operates in the *data space*, we extend the DRO framework to the *weight space* and interpret model merging as a form of distributional uncertainty over model parameters.

## 3.2 MERGOPT: A MERGE-AWARE OPTIMIZER VIA WEIGHT ROBUST OPTIMIZATION

In this section, we introduce our proposed method, MergOPT, which aims to enhance the robustness of model merging through weight-space robust optimization. More specifically, we treat the merging process as a form of merge offsets in the weight space and apply distributionally robust optimization techniques in the fine-tuning stage to train models that are resilient to various merging scenarios.

### 3.2.1 REFORMULATING MODEL MERGING AS WEIGHT-SPACE MERGE OFFSETS

Consider fine-tuning on task $k$, where the resulting model parameters can be expressed as $\theta_k = \theta_0 + \Delta\theta_k$, where $\Delta\theta_k$ is the task vector corresponding to task $k$. When merging $K$ tasks, the merged model parameters $\theta_{\text{merged}} = \phi(\theta_k, \zeta(\alpha, K, \Delta\theta))$ can be reformulated as:

$$\phi(\theta_k, \zeta(\alpha, K, \Delta\theta)) := \theta_0 + \alpha \sum_{j=1}^K \Delta\theta_j = (\theta_0 + \Delta\theta_k) - \Delta\theta_k + \alpha \sum_{j=1}^K \Delta\theta_j = \theta_k + \underbrace{\left( (\alpha - 1)\Delta\theta_k + \alpha \sum_{j \neq k} \Delta\theta_j \right)}_{\zeta(\alpha, K, \Delta\theta)}, \tag{5}$$

where $\phi(\theta_k, \zeta(\alpha, K, \Delta\theta))$ formalizes the process of merging the current task-specific model (i.e., $\theta_k$) with the remaining fine-tuned models. The additional term $\zeta(\alpha, K, \Delta\theta)$ represents the parameter offset introduced by the merging operation, which depends on the merging coefficient $\alpha$, the number of tasks $K$, and the task vectors $\Delta\theta_j$ ($j \in \{1, 2, \ldots, K\}$) from the other models. It is worth emphasizing that this formulation is consistent with SAFT-Merge (Lee et al., 2025), but our interpretation and solution strategy are fundamentally different. We provide a detailed comparison between the two methods in both the related work section and our experiments.

### 3.2.2 WEIGHT ROBUST OPTIMIZATION OBJECTIVE

Building on the above interpretation of model merging as merge-induced parameter shifts in weight space, we argue that a merge-aware optimizer during fine-tuning should satisfy two key objectives: ❶ Preservation Objective: Preserve the standard task loss $\mathcal{L}_{\text{task}}(\theta_k; \mathcal{D}_k)$ to ensure strong performance on the current task. ❷ Robustness Objective: Enhance robustness to diverse merging scenarios by accounting for the worst-case merging parameters $\zeta(\alpha, K, \Delta\theta)$ within a feasible set $\mathcal{B}$.

Formally, we define the weight-space robust optimization (WRO) objective as:

$$\min_{\theta_k} \sup_{(\alpha, K, \Delta\theta) \in \mathcal{B}} \mathbb{E}\Big[\ell_k\big(\phi(\theta_k, \zeta(\alpha, K, \Delta\theta))\big)\Big], \tag{6}$$

where $\phi(\theta_k, \zeta)$ denotes the merged parameters under merge-induced offset $\zeta$, and $\mathcal{B}$ is the ambiguity set in the weight space, capturing feasible merging configurations. The feasible set is defined as

$$\mathcal{B} = \Big\{(\alpha, K, \Delta\theta): \ \alpha \in \mathcal{A}, \ K \in \mathbb{Z}_{>0}, \ K \leq K_{\max}, \ \Delta\theta \in \mathcal{Z} \subseteq \text{span}\{\Delta\theta_1, \Delta\theta_2, \ldots, \Delta\theta_K\}\Big\}, \tag{7}$$

which constrains the merging configuration so that the induced merge offsets remain within reasonable bounds. Here, $\mathcal{A}$ denotes the set of admissible merging coefficients, $K_{\max}$ specifies the maximum number of tasks allowed for merging, and $\mathcal{Z}$ is the set of admissible merge-offset vectors, restricted to linear combinations of task-specific parameter differences $\Delta\theta_j$. Throughout, $\text{span}(\cdot)$ denotes the linear span of a given set of task vectors. Naturally, when Eq. 6 reaches its optimal solution, $\theta_k$ inherently satisfies both the previously introduced preservation objective and robustness objective.

To solve the proposed WRO objective in Eq. 6, we can employ an alternating optimization strategy. The outer minimization updates the model parameters $\theta_k$ using gradient descent, while the inner maximization finds the worst-case merging parameters $\zeta(\alpha, K, \Delta\theta)$ using projected gradient ascent. Specifically, the optimization proceeds as follows: ❶ *Inner Maximization:* Given the current task parameters $\theta_k$, identify the worst-case merging configuration within the feasible set $\mathcal{B}$. The adversarial merging parameters are then obtained by solving the following problem:

$$(\alpha^*, K^*, \Delta\theta^*) = \text{Proj}_{\mathcal{B}}(\alpha', K', \Delta\theta') = \arg \max_{(\alpha', K', \Delta\theta')} \mathcal{L}_{\text{task}}\big(\phi(\theta_k^*, \zeta(\alpha', K', \Delta\theta')); \mathcal{D}_k\big). \tag{8}$$

where $\text{Proj}_{\mathcal{B}}(x)$ is a projection operator that projects the $x$ onto the feasible set $\mathcal{B}$. ❷ *Outer Minimization:* Update $\theta_k$ to minimize the loss under the adversarially selected merging parameters. This ensures that the model adapts to the worst-case weight-shift introduced by the merging operation:

$$\theta_k^* \ \leftarrow \ \theta_k - \eta \, \nabla_{\theta_k}\Big(\mathcal{L}_{\text{task}}\big(\phi(\theta_k, \zeta(\alpha^*, K^*, \Delta\theta^*)); \mathcal{D}_k\big)\Big), \tag{9}$$

where $\eta$ is the learning rate.

This alternating minimax procedure continues until convergence, effectively ensuring that the fine-tuned model $\theta_k$ is not only optimized for its own task but also robust to a broad range of potential merging scenarios. This alternating optimization strategy follows classical machine learning frameworks like gradient ascent/descent and min–max optimization (Boyd & Vandenberghe, 2004). However, for model merging, the objective is impractical due to inaccessible task vectors and costly inner optimization. We tailor this framework to merging with efficient approximations in next setion.

### 3.2.3 OPTIMIZATION STRATEGY

In this subsection, we discuss the practical challenges associated with optimizing the WRO objective in Eq. 6 and propose strategies to address these challenges.

**Optimization Challenges.** From the optimization problem in Sec. 3.2.2, the ideal worst-case robust optimization faces the following two constraints during practical optimization, rendering it inapplicable in real-world scenarios: ❶ *Unknowable Merge-Offset Variables*: As derived from Eq. 5 and Eq. 7, the merge-induced parameter shift $\zeta$ is primarily determined by three types of variables: the merging coefficient $\alpha$, the number of merged tasks $K$, and the task vectors $\Delta\theta = \{\Delta\theta_1, \ldots, \Delta\theta_K\}$. However, when fine-tuning the model $\theta_k$ for task $k$ (or its corresponding task vector $\Delta\theta_k$), the task vectors of other models to be merged are often inaccessible—this is because different developers typically fine-tune their respective models independently. Under such circumstances, the merging coefficient and the number of merged tasks that can achieve optimal merging performance are

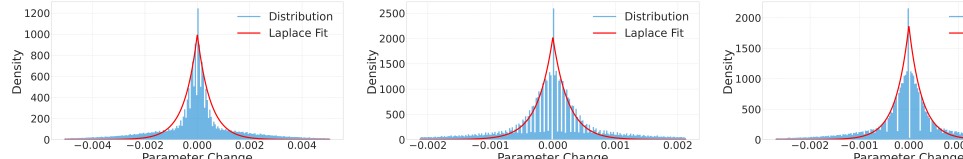

Figure 1: Distribution of parameter changes (i.e., task vectors) after fine-tuning different large language models: from left to right are Llama-3.2-1B, Qwen2.5-1.5B, and Llama-3.2-3B. The blue curve represents the empirical distribution, while the red curve shows the fitted Laplace distribution. We observe that the Laplace distribution provides a close approximation to the empirical distribution.

naturally unascertainable in advance. ❷ *Inefficient Inner Optimization*: Even though some of the aforementioned variables have been clearly defined, the solution process for the worst-case merge offset described in Eq. 8 remains highly time-consuming. Specifically, the merging coefficients $\alpha$, the number of merged tasks $K$, and the task-vector space $\mathcal{Z}$ together induce a feasible set whose size grows exponentially. Performing an explicit worst-case maximization over this space would be computationally intractable, both in theory and in practice.

**Feasible Set Approximation.** To address the issue of unknowable variables, we propose to effectively approximate the feasible set based on a series of prior information. More specifically, we make the following approximations. ❶ *For task vectors* $\{\theta_i\}_{i=1}^K$, we conducted analysis across three large language model architectures and seven downstream tasks (detailed in Sec. 4.1). We visualized the distribution of the cumulative task vectors (i.e., $\sum_{i=1}^K \Delta\theta_i$) in the main text and individual task vectors (i.e., $\{\Delta\theta_i\}_{i=1}^K$) in Appendix D.1.1. As shown in Fig. 1, all task vectors can be well-fitted by a specific Laplace distribution (Kotz et al., 2012), where an overwhelming majority of elements are concentrated around 0. ❷ *For the optimal merging coefficient* $\alpha$, prior studies on model merging consistently indicate that $\alpha$ typically lies in the interval $(0, 1)$. Since fine-grained parameter search is computationally expensive, most works adopt a fixed small value; for example, $\alpha = 0.3$ is commonly used in Task Arithmetic (Ilharco et al., 2023) and Ties-Merging (Yadav et al., 2023). To further validate this observation, we evaluated the impact of different $\alpha$ values on merging performance (see Tab. 13 in Appendix D.1.2), which provides a practical discrete candidate set for $\alpha$. ❸ *For the maximum number of models to be merged*, existing evidence shows that performance degrades more severely as more models are merged (Yadav et al., 2024). Consequently, most experiments restrict the number of merged models to fewer than ten. In the context of LLMs, merging is typically limited to two or three models (Goddard et al., 2024; Wan et al., 2024b; Yu et al., 2024; Du et al., 2024; Akiba et al., 2025), and few works explore merging at much larger scales (Wang et al., 2025)..

**Single-Step Merge-Offset Approximation.** To alleviate the computational inefficiency of iterative inner maximization, we approximate Eq. 8 using a single-step sampling strategy over merging configurations. Instead of performing multiple projection gradient-based updates to identify the worst-case merging parameters, we directly sample $(\alpha, K, z)$ from their respective feasible distributions and construct the merge-offset model in one step. In particular, $z$ is drawn from a Laplace distribution fitted to the empirical task vectors, as established in the previous analysis. The offsets are then given by $\phi(\theta_k, \zeta(\alpha, K, z))$, and $\zeta(\alpha, K, z) = (K\alpha - 1)z$ denotes the merge-induced parameter offset induced by merging $K$ tasks with coefficient $\alpha$. To save sampling time, we note that each task vector is assumed to be $z$ here. This single-step approximation substantially reduces the computational overhead while retaining the essential characteristics of the worst-case merge offsets. Moreover, since the offset $z$ is sampled from a Laplace distribution that matches the empirical distribution of task vectors, repeated sampling naturally increases the probability of capturing directions that are close to the true worst-case, or at least adversarially challenging merge offsets. This allows us to achieve substantial robustness improvements even when the combinatorial space is extremely large and computing the theoretical optimum is infeasible.

**Final Practical Objective.** By combining the feasible-set approximation with the single-step merge-offset strategy, we derive a practical optimization objective that can be efficiently implemented during fine-tuning. At each training step, we first sample the merging coefficient $\alpha$ from the discrete candidate set $\mathcal{A}$, the number of tasks $K$ from $\{1, 2, \ldots, K_{\max}\}$, and the offset vector $z$ from a Laplace distribution fitted to the empirical task vectors. The merge-offset model parameters are then obtained as $\phi(\theta_k, \zeta(\alpha, K, z))$. The task loss is evaluated at these parameters and used to update $\theta_k$. Formally,

Table 1: Performance comparison of model merging methods with Llama-3.2-1B-Instruct.

| Method | Task Performance | | | | | | | Avg. (↑) |
|---|---|---|---|---|---|---|---|---|
| | C-STANCE | FOMC | MeetingBank | ScienceQA | NumGLUE-cm | NumGLUE-ds | 20Minuten | |
| Pre-Trained | 0.3386 | 0.2581 | 0.2036 | 0.6780 | 0.1220 | 0.1646 | 0.3802 | 0.3064 |
| Standard Fine-Tuned | 0.4980 | 0.5988 | 0.3707 | 0.8524 | 0.3902 | 0.5793 | 0.3880 | 0.5254 |
| MergOPT Fine-Tuned | 0.4957 | 0.6331 | 0.3158 | 0.8780 | 0.4390 | 0.5305 | 0.3829 | 0.5250 |
| Weight Averaging | 0.4206 | 0.5040 | 0.2179 | 0.7340 | 0.2195 | 0.3171 | 0.3813 | 0.3992 |
| **Weight Averaging w/ MergOPT** | 0.4202 | 0.4536 | 0.2093 | 0.7555 | 0.2927 | 0.3720 | 0.3825 | 0.4123₍₊₃.₂₈%₎ |
| Task Arithmetic | 0.4219 | 0.4980 | 0.2094 | 0.7370 | 0.2439 | 0.3476 | 0.3805 | 0.4055 |
| **Task Arithmetic w/ MergOPT** | 0.4203 | 0.4718 | 0.2077 | 0.7530 | 0.3171 | 0.3659 | 0.3797 | 0.4165₍₊₂.₇₁%₎ |
| TIES-Merging | 0.4236 | 0.4738 | 0.2145 | 0.7430 | 0.2439 | 0.3537 | 0.3862 | 0.4055 |
| **TIES-Merging w/ MergOPT** | 0.4202 | 0.4536 | 0.2093 | 0.7555 | 0.2927 | 0.3720 | 0.3825 | 0.4123₍₊₁.₆₈%₎ |
| DARE | 0.4143 | 0.3810 | 0.2120 | 0.7180 | 0.2195 | 0.3232 | 0.3821 | 0.3786 |
| **DARE w/ MergOPT** | 0.4192 | 0.4819 | 0.2107 | 0.7485 | 0.2927 | 0.3659 | 0.3843 | 0.4147₍₊₉.₅₄%₎ |

the practical training objective at each step can be expressed as:

$$\min_{\theta_k} \ \mathbb{E}_{\alpha,K,z} \left[ \mathcal{L}_{\text{task}}\big(\phi(\theta_k, \zeta(\alpha, K, z)); \mathcal{D}_k\big) \right],$$
$$\text{s.t. } \alpha \sim \text{Uniform}(\mathcal{A}), \quad K \sim \text{Uniform}(\{1, 2, \ldots, K_{\max}\}), \quad z \sim \text{Laplace}(\mu, b),$$

$$(10)$$

where the expectation is taken over the sampled merging parameters $(\alpha, K, z)$. The Laplace distribution is defined as $\text{Laplace}(\mu, b) = \frac{1}{2b} \exp\left(-\frac{|x-\mu|}{b}\right)$, with location parameter $\mu$ and scale parameter $b$. The model parameters are finally updated using stochastic gradient descent: $\theta_k \leftarrow \theta_k - \eta \nabla \mathcal{L}_{\text{task}}\big(\phi(\theta_k, \zeta); \mathcal{D}_k\big)$. The optimization procedure is summarized in Alg. 1 of Appendix B.3.

# 4 EXPERIMENT

## 4.1 EXPERIMENTAL SETUP

In this section, we detail the experimental setup used to evaluate the effectiveness of our proposed method. Due to space limitations, more experimental details can be found in Appendix B.

**Datasets and Metrics.** We evaluate our proposed method on seven datasets from TraceBench (Wang et al., 2023), including C-STANCE (Zhao et al., 2023), FOMC (Shah et al., 2023), MeetingBank (Hu et al., 2023), ScienceQA (Lu et al., 2022), NumGLUE-cm (Mishra et al., 2022), NumGLUE-ds (Mishra et al., 2022), and 20Minuten (Rios et al., 2021). These datasets span a variety of tasks, including domain-specific applications, multilingual understanding, and mathematical reasoning. The evaluation metric for each task is as follows: accuracy for C-STANCE, FOMC, ScienceQA, NumGLUE-cm, and NumGLUE-ds; ROUGE-L for MeetingBank; and SARI for 20Minuten. For all metrics, the higher the value, the better. We report the average score across all tasks as the overall performance metric. The statistics of these datasets are summarized in Table 6 in the Appendix.

**Base Models and Optimizers.** We conduct experiments on four base models: Llama-3.2-1B-Instruct (Meta, 2024), Qwen2.5-1.5B-Instruct (Qwen et al., 2025), Llama-3.2-3B-Instruct (Meta, 2024), and Llama-3.1-8B-Instruct (Meta, 2024). In this work, we adopt AdamW (Loshchilov & Hutter, 2017) as the default base optimizer for both the standard fine-tuning baseline and our merge-aware fine-tuning; implementation details are provided in Appendix B.2. We further verify optimizer agnosticism by instantiating our method with SGD and by comparing it against the SAM optimizer; the corresponding results are reported in Appendix D.2.

**Merging Methods.** Since our method operates during the fine-tuning stage, it remains independent of the specific choice of merging algorithms. To verify its effectiveness, we employ four representative merging strategies, including Weight Averaging (Wortsman et al., 2022), Task Arithmetic (Ilharco et al., 2023), TIES-Merging (Yadav et al., 2023), and DARE (Yu et al., 2024), applied to models obtained from both standard fine-tuning and our proposed MergOPT fine-tuning approach.

## 4.2 PERFORMANCE COMPARISON AND ANALYSIS

This section presents the main results and analysis of our experiments, demonstrating the effectiveness of MergOPT in enhancing merging performance. More experimental results and analyses can be found in Appendix C and Appendix D.

**Robustness Across Architectures and Downstream Tasks.** In Tables 1 and 2, we evaluate robustness under the challenging setting of merging seven independently fine-tuned expert models,

Table 2: Performance comparison of model merging methods with Llama-3.2-3B-Instruct.

| Method | Task Performance | | | | | | | Avg. (↑) |
|---|---|---|---|---|---|---|---|---|
| | C-STANCE | FOMC | MeetingBank | ScienceQA | NumGLUE-cm | NumGLUE-ds | 20Minuten | |
| Pre-Trained | 0.4082 | 0.3528 | 0.2054 | 0.8962 | 0.1707 | 0.2195 | 0.3857 | 0.3770 |
| Standard Fine-Tuned | 0.5415 | 0.6835 | 0.4317 | 0.9335 | 0.6098 | 0.6463 | 0.3898 | 0.6057 |
| `MergOPT` Fine-Tuned | 0.5545 | 0.6653 | 0.3896 | 0.9360 | 0.5122 | 0.6341 | 0.3886 | 0.5836 |
| Weight Averaging | 0.4617 | 0.5665 | 0.2213 | 0.9140 | 0.4390 | 0.4146 | 0.3891 | 0.4866 |
| **Weight Averaging w/ `MergOPT`** | 0.4700 | 0.5867 | 0.2181 | 0.9180 | 0.4390 | 0.4085 | 0.3876 | 0.4897 (+0.64%) |
| Task Arithmetic | 0.4685 | 0.5605 | 0.2186 | 0.9100 | 0.4634 | 0.4024 | 0.3862 | 0.4871 |
| **Task Arithmetic w/ `MergOPT`** | 0.4755 | 0.5948 | 0.2167 | 0.9110 | 0.4878 | 0.4573 | 0.3883 | 0.5045 (+3.56%) |
| TIES-Merging | 0.4670 | 0.5706 | 0.2217 | 0.9130 | 0.4634 | 0.4024 | 0.3906 | 0.4898 |
| **TIES-Merging w/ `MergOPT`** | 0.4770 | 0.6028 | 0.2142 | 0.9175 | 0.5122 | 0.4573 | 0.3878 | 0.5098 (+4.09%) |
| DARE | 0.4630 | 0.5867 | 0.2203 | 0.9055 | 0.4634 | 0.4085 | 0.3871 | 0.4906 |
| **DARE w/ `MergOPT`** | 0.4690 | 0.6129 | 0.2198 | 0.9140 | 0.4878 | 0.4451 | 0.3851 | 0.5048 (+2.89%) |

Table 3: Performance of different merging methods on 4-task groups (Llama-3.2-1B-Instruct).

| Method | Group 1 | | | | | Group 2 | | | | |
|---|---|---|---|---|---|---|---|---|---|---|
| | FOMC | MeetingBank | NumGLUE-ds | 20Minuten | Avg. (↑) | C-STANCE | FOMC | ScienceQA | NumGLUE-cm | Avg. (↑) |
| Task Arithmetic | 0.4093 | 0.2412 | 0.4512 | 0.3816 | 0.3708 | 0.4320 | 0.4637 | 0.7710 | 0.3171 | 0.4959 |
| **Task Arithmetic w/ `MergOPT`** | 0.4758 | 0.2320 | 0.4512 | 0.3814 | 0.3851 (+3.86%) | 0.4320 | 0.4597 | 0.7945 | 0.3659 | 0.5130 (+3.45%) |
| TIES-Merging | 0.3992 | 0.2412 | 0.4573 | 0.3846 | 0.3706 | 0.4285 | 0.4718 | 0.7790 | 0.3415 | 0.5052 |
| **TIES-Merging w/ `MergOPT`** | 0.4657 | 0.2352 | 0.4512 | 0.3843 | 0.3841 (+3.64%) | 0.4274 | 0.4698 | 0.7985 | 0.3659 | 0.5154 (+2.02%) |

comparing the performance of different merging strategies using Llama-3.2-1B and Llama-3.2-3B as the base models, respectively. We assess robustness by applying four representative merging strategies (Weight Averaging, Task Arithmetic, TIES-Merging, and DARE) to models fine-tuned either with the standard procedure or with `MergOPT`. Across all model scales, incorporating `MergOPT` consistently improves the merged performance. For example, in Table 1, Weight Averaging combined with `MergOPT` achieves an average score of 0.4123, outperforming plain Weight Averaging (0.3992). The same trend is observed for Task Arithmetic (0.4165 vs. 0.4055) and DARE (0.4147 vs. 0.3786). Then, on the larger Llama-3.2-3B (Table 2), the merged models with `MergOPT` still show consistent improvements, e.g., Task Arithmetic increases from 0.4871 to 0.5045 (+3.6%) and TIES-Merging from 0.4898 to 0.5098 (+4.1%). On average, these enhancements correspond to about 3.5% relative improvement across 8 cases, with the largest observed gain reaching 9.5% (i.e., DARE on Llama-3.2-1B). Tables 7 and 8 in the appendix further demonstrate that we have validated the effectiveness of our method on Qwen2.5-1.5B and LLama-8B. In a word, the improvements are consistent across model scales and merging methods, highlighting the generality and practicality of our approach.

**Robustness to the Number of Tasks.** Tables 1–2 report results when merging all seven expert models. To further evaluate the effectiveness of our method under varying numbers of tasks, we also conduct experiments on smaller groups of experts, specifically 2-task, 4-task, and 6-task settings. Due to space constraints, Table 3 presents results on 4-task groups, while the results for 2-task and 6-task groups are deferred to the Appendix (Tables 20 and 21). Concretely, we randomly sample two groups of four tasks from the full set of seven, merge the corresponding fine-tuned models within each group, and then evaluate the merged models. As shown in Table 3, Task Arithmetic w/ `MergOPT` achieves an average score of 0.3851 in Group 1, compared to 0.3708 without `MergOPT` yielding a 3.86% relative improvement. Similarly, in Group 2 the average score improves from 0.4959 to 0.5130 (+3.45%). Comparable gains are also observed for TIES-Merging, highlighting that our method consistently enhances merging robustness and generality across different task configurations.

**Robustness to Merging Coefficients.** In this part, we visualize the joint loss landscapes of models fine-tuned with the standard AdamW optimizer and with our proposed `MergOPT` method, in order to illustrate the robustness of our approach under varying merging coefficients. Specifically, we randomly selected four pairs of tasks (e.g., C-STANCE & MeetingBank, MeetingBank & ScienceQA), and plotted contour maps of the joint-task loss as a function of the merging coefficients. As shown in Figure 2, each pixel in the heatmap corresponds to the joint loss value of a merged model defined by $\theta_{\text{merged}} = \theta_0 + \alpha_1 \Delta\theta_1 + \alpha_2 \Delta\theta_2$, where the joint loss is given by $L(\theta_{\text{merged}}; \mathcal{D}_1) + L(\theta_{\text{merged}}; \mathcal{D}_2)$. The horizontal and vertical axes represent the merging coefficients $(\alpha_1, \alpha_2)$, while the color intensity indicates the magnitude of the loss. Across the four task pairs, we observe the following: (i) *AdamW fine-tuning* (left column): the low-loss regions are relatively narrow, and the loss increases sharply as the merging coefficients deviate from the optimum. This indicates sensitivity to merging shifts and weaker robustness. (ii) *Our method* (right column): the low-loss regions are substantially larger, and the contours around the optimum are flatter. This suggests that models fine-tuned with our optimizer exhibit greater stability under merging shifts, allowing them to better tolerate diverse coefficient configurations. These visualizations provide intuitive evidence that our method leads to more favorable loss landscapes for model merging.

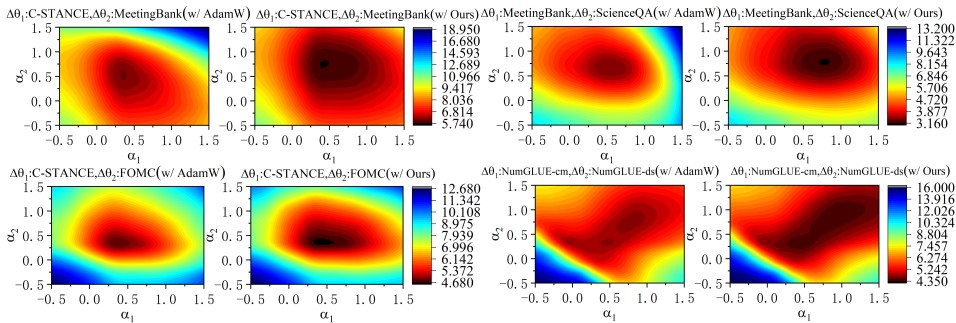

Figure 2: Visualization of the joint loss landscape for two task-specific models across two downstream tasks. The left panel shows AdamW-fine-tuned models, the right shows models fine-tuned with our `MergOPT`; darker colors indicate lower loss.

## 4.3 COMPARE WITH OTHER FINE-TUNING METHODS

**Compare with SAM-based Fine-Tuning**. SAFT-Merge (Lee et al., 2025) employs SAM-based optimizers (Foret et al., 2021; Kwon et al., 2021) during the fine-tuning stage to improve model mergeability. We compare SAFT-Merge and `MergOPT` from two perspectives: (i) As shown in Table 4 (a), under the same number of training epochs, SAFT-Merge and MergOPT each exhibit distinct advantages. For instance, under Weight Averaging and DARE, `MergOPT` outperforms SAFT-Merge (SAM) by 2.59% and 5.20%, respectively. In contrast, under Task Arithmetic and TIES-Merging, `MergOPT` falls behind by 2.23% and 1.29%. However, SAM-based optimization requires 2.04× the cost of AdamW, whereas `MergOPT` incurs only a 1.17× cost. (ii) Furthermore, Table 4 (b) demonstrates that when training time is comparable, `MergOPT` consistently surpasses SAFT-Merge. More specifically, `MergOPT` achieves improvements of 4.08%, 0.70%, 0.95%, and 3.99% over SAFT-Merge across the four merging strategies. In addition, comparing SAFT-Merge (SAM) and SAFT-Merge (ASAM), we observe that ASAM is generally superior when trained for the same number of epochs, whereas SAM becomes slightly better under comparable training cost. Under the same configuration, the two methods exhibit similar training time.

Table 4: Performance comparison of `MergOPT` and SAFT-Merge (based on SAM and ASAM) under (a) equal training epochs or (b) close to the training duration on Llama-3.2-1B-Instruct.

| Method | Weight Averaging | Task Arithmetic | TIES-Merging | DARE | Time |
|---|---|---|---|---|---|
| (a) SAFT-Merge (SAM) | 0.3980 | 0.4260 | 0.4177 | 0.3942 | 2.04× |
| (a) SAFT-Merge (ASAM) | 0.4047(+1.68%) | 0.4119(-3.31%) | 0.4226(+1.17%) | 0.4101(+4.03%) | 2.06× |
| **(a) MergOPT** | **0.4083**(+2.59%) | **0.4165**(-2.23%) | **0.4123**(-1.29%) | **0.4147**(+5.20%) | 1.17× |
| (b) SAFT-Merge (SAM) | 0.3923 | 0.4136 | 0.4084 | 0.3988 | 1.04× |
| (b) SAFT-Merge (ASAM) | 0.3868(-1.40%) | 0.4181(+1.09%) | 0.4030(-1.32%) | 0.3943(-1.13%) | 1.04× |
| **(b) MergOPT** | **0.4083**(+4.08%) | **0.4165**(+0.70%) | **0.4123**(+0.95%) | **0.4147**(+3.99%) | 1.17× |

**Compare with Tangent Space Fine-Tuning**. Ortiz-Jimenez et al. (2023) proposes fine-tuning models in the tangent space to enhance their mergeability. In Table 5, we compare standard fine-tuning, tangent-space fine-tuning, and our `MergOPT` fine-tuning. We observe that: (i) under Task Arithmetic, TIES-Merging, and DARE, tangent-space fine-tuning improves over standard fine-tuning by 3.14%, 3.95%, and 2.54%, respectively, but drops by 8.01% under Weight Averaging; (ii) `MergOPT` improves over standard fine-tuning by 2.37%, 3.44%, 4.10%, and 3.58% under Weight Averaging, Task Arithmetic, TIES-Merging, and DARE, respectively, and yields larger gains than tangent-space fine-tuning. Note that the inference cost of linear models obtained via tangent-space fine-tuning is typically 2-3× that of standard models, leading to efficiency issues.

Table 5: Performance comparison of `MergOPT` and Tangent Space Fine-tuning on ViT-B/32.

| Method | Weight Averaging | Task Arithmetic | TIES-Merging | DARE | WUDI-Merging |
|---|---|---|---|---|---|
| Standard | 54.9 | 66.9 | 65.8 | 67.0 | 83.2 |
| Tangent | 50.5(-8.01%) | 69.0(+3.14%) | 68.4(+3.95%) | 68.7(+2.54%) | 77.1(-7.33%) |
| **MergOPT** (Ours) | **56.2**(+2.37%) | **69.2**(+3.44%) | **68.5**(+4.10%) | **69.4**(+3.58%) | **84.3**(+1.32%) |

**Summary.** These results indicate that `MergOPT` is both more efficient and more effective than SAM-based and tangent-space fine-tuning methods.

## 5 CONCLUSION AND FUTURE WORK

This paper introduced a novel fine-tuning optimizer (`MergOPT`) designed to enhance the robustness of expert models during model merging. By reformulating fine-tuning as a robust optimization problem in the weight space, our method guides models to converge toward minima that are more amenable to merging and more resilient to the parameter changes introduced at the merging stage. Extensive experiments demonstrate that `MergOPT` consistently improves the performance of merged models. Several promising avenues for future research remain: First, developing more accurate approximation techniques to simulate key merging factors during fine-tuning could further enhance the model's adaptability to merging. Second, combining `MergOPT` with other robustness-oriented training techniques may further strengthen merging stability. Third, integrating the models trained via our `MergOPT` method into various more advanced merging schemes is also feasible and valuable.

### ACKNOWLEDGMENTS

This research is supported by the National Natural Science Foundation of China (Grant Nos. 62576364, 62576083, 62411540034, U2541229), the Shenzhen Basic Research Key Project (Natural Science Foundation) (No. JCYJ20241202124430041), the China Postdoctoral Science Foundation under Grant Number 2025M781535, the Open Research Fund from Guangdong Laboratory of Artificial Intelligence and Digital Economy (SZ) (NO. GML-KF-24-23), and Ningbo Science and Technology Innovation 2025 Major Project (2025Z027).

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

**Appendix Contents.** The appendix is structured into several sections, each presenting supplementary information and detailed explanations to support the main text.

## A  LLM USAGE STATEMENT

This paper makes use of a large language model (ChatGPT) exclusively for language polishing, spelling correction, and grammar checking. The LLM was not involved in literature retrieval or in the development of specific ideas. Following the polishing process, the authors carefully reviewed and revised the content as necessary and assume full responsibility for the final published version.

## B  EXPERIMENTAL DETAILS

In this section, we provide detailed statistics of the datasets used in our experiments (Sec. B.1) and elaborate on the implementation details of our proposed `MergOPT` method (Sec. B.2).

Table 6: An overview of dataset statistics in experiments. 'Source' indicates the origin of the context. 'Avg. Len.' denotes the average length in words for English, German, and code datasets, and in characters for Chinese. 'SARI' is a metric specific to simplification. For all metrics, the larger the corresponding value, the better.

| Dataset | Source | Language | Avg. Len. | Metric |
|---|---|---|---|---|
| *Domain-specific* | | | | |
| ScienceQA (Lu et al., 2022) | Science | English | 210 | Accuracy |
| FOMC (Shah et al., 2023) | Finance | English | 51 | Accuracy |
| MeetingBank (Hu et al., 2023) | Meeting | English | 2,853 | ROUGE-L |
| *Multi-lingual* | | | | |
| C-STANCE (Zhao et al., 2023) | Social media | Chinese | 127 | Accuracy |
| 20Minuten (Rios et al., 2021) | News | Germany | 382 | SARI |
| *Mathematical reasoning* | | | | |
| NumGLUE-cm (Mishra et al., 2022) | Math | English | 32 | Accuracy |
| NumGLUE-ds (Mishra et al., 2022) | Math | English | 21 | Accuracy |

## B.1 DATASET STATISTICS

This section provides detailed statistics of the datasets used in our experiments, as summarized in Table 6. The datasets in TraceBench (Wang et al., 2023) are constructed based on the following principles: (i) they are sufficiently novel such that most LLMs have not been trained on them; (ii) they are designed to pose a meaningful level of challenge to LLMs; and (iii) they cover a diverse range of tasks to provide a comprehensive evaluation of model capabilities. A detailed description of the seven datasets is provided below.

**Domain-specific Applications.** ScienceQA (Lu et al., 2022) is a multi-hop question answering dataset built upon elementary and high school science curricula. It exhibits rich domain diversity, covering natural sciences, social sciences, and language sciences. FOMC (Shah et al., 2023) is a novel financial-domain classification task focused on hawkish–dovish categorization. The dataset consists of three subsets: meeting minutes, press conference transcripts, and speeches, each capturing different aspects of monetary policy communication. MeetingBank (Hu et al., 2023) is a new benchmark dataset for summarization of city council meetings. It requires a comprehensive understanding of lengthy background materials, making it particularly challenging.

**Multilingual Understanding Tasks.** C-STANCE (Zhao et al., 2023) is a zero-shot stance detection dataset collected from Sina Weibo, one of the most popular social media platforms in China. It serves as a benchmark for evaluating models' ability to understand and analyze Chinese text. 20Minuten (Rios et al., 2021) is a text simplification dataset consisting of full-length articles paired with shorter, simplified summaries from a Swiss news magazine. It provides a benchmark for assessing models' capability in generating German text, particularly for simplification tasks.

**Mathematical Reasoning Tasks.** NumGLUE (Mishra et al., 2022) is designed to evaluate the mathematical reasoning ability of AI systems, with a core focus on understanding and performing basic arithmetic. In our experiments, we adopt two subsets: NumGLUE-cm (Commonsense), which involves simple arithmetic computations based on mathematical facts, and NumGLUE-ds (Domain Specific), which extends arithmetic reasoning by requiring additional domain-specific knowledge.

## B.2 IMPLEMENTATION DETAILS

The main experiments in this work focus on LLM architectures and language tasks, while the details for vision tasks are provided separately in Section C.3. More specifically, we build our experiments on the HuggingFace Transformers (Wolf et al., 2020) library for loading pre-trained models and conducting task-specific fine-tuning. The pre-trained models include Llama-3.2-1B-Instruct (Meta, 2024), Qwen2.5-1.5B-Instruct (Qwen et al., 2025), Llama-3.2-3B-Instruct (Meta, 2024), and Llama-3.1-8B-Instruct (Meta, 2024). Unless otherwise specified, we adopt AdamW (Loshchilov & Hutter, 2017) as the default base optimizer during fine-tuning. Following the TraceBench (Wang et al., 2023) protocol, we fine-tune the models on C-STANCE, FOMC, MeetingBank, ScienceQA, NumGLUE-cm,

---

**Algorithm 1** `MergOPT`: A Merge-Aware Optimizer for Robust Model Merging

---

**Require:** Pretrained model $f_{\theta_0}$, task dataset $\mathcal{D}_k$, candidate merging coefficient set $\mathcal{A}$, maximum number of merged tasks $K_{\max}$, Laplace distribution parameters $(\mu, b)$, base optimizer (e.g., SGD or AdamW)
**Ensure:** Fine-tuned parameters $\theta_k$
1: Initialize $\theta_k \leftarrow \theta_0$
2: **for** each training step **do**
3:     Sample a mini-batch $\mathcal{B}_k \leftarrow \{(x_i, y_i)\}_{i=1}^{|\mathcal{B}_k|} \sim \mathcal{D}_k$
4:     Sample merging parameters: $\alpha \sim \text{Uniform}(\mathcal{A})$, $\quad K \sim \text{Uniform}(\{1, 2, \ldots, K_{\max}\})$, $\quad z \sim \text{Laplace}(\mu, b)$
5:     Construct merge-offset parameters: $\theta_k' \leftarrow \phi(\theta_k, \zeta(\alpha, K, z)) = \theta_k + (K\alpha - 1)z$
6:     Compute task loss and gradient at $\theta_k'$: $g \leftarrow \nabla \mathcal{L}_{\text{task}}(\theta_k'; \mathcal{B}_k)$
7:     Update parameters using the base optimizer: $\theta_k \leftarrow \text{Optimizer}(\theta_k, g)$
8: **end for**
9: **return** $\theta_k$

---

NumGLUE-ds, and 20Minuten for 5, 3, 7, 3, 5, 5, and 7 epochs, respectively. The learning rate is set to 2e-5, the batch size to 8, and the weight decay to 0.001.

For our proposed `MergOPT` method, we set the default parameters of the Laplace distribution to $(\mu, b) = (0, 0.0005)$. We further evaluate different values of $b$, including 0.05, 0.001, and 0.0005 in Table 11, and observe consistently strong robustness across these settings. For the feasible set $\mathcal{A}$ of merging coefficients, we adopt a default configuration of $\mathcal{A} = [0.1, 0.2, 0.3, 0.4, 0.5, 0.6]$. Additionally, we report results in Table 13 for coefficient values ranging from 0.1 to 1.0 with an increment of 0.1. The maximum number of tasks $K_{\max}$ considered for merging is set to 7 by default.

All experiments are conducted on a machine with an Intel(R) Xeon(R) Gold 6459C CPU (12 cores), NVIDIA RTX 4090 GPUs (48 GB memory), and 90 GB of RAM. The software environment consists of Python 3.8 and PyTorch 2.1.2.

### B.3 ALGORITHM

The pseudocode of the proposed MergOPT algorithm is presented in Algorithm 1. Given the input hyperparameters, for each task ($k$) we initialize the model with the pretrained parameters (Line 1). At each optimization step, we first sample a mini-batch of data (Line 3), then sample the merging coefficients, the number of tasks, and the task vectors from a predefined feasible region (Line 4). Next, we construct a merged parameter offset (Line 5), and finally compute the gradients (Line 6) and update the parameters (Line 7).

## C ADDITIONAL EXPERIMENTAL RESULTS

In this section, we validate the effectiveness of the proposed method on other architectures, including two LLMs (Qwen2.5-1.5B-Instruct and Llama3.1-8B-Instruct) and a vision model (ViT-B/32).

### C.1 APPLICATIONS IN OTHER LLM ARCHITECTURE

Beyond the mainstream Llama (e.g., Tables 1 and 2) architecture, we also validate the effectiveness of our proposed method on the Qwen architecture by merging seven Qwen2.5-1.5B-Instruct models, each fine-tuned separately on a single task. As shown in Table 7, our `MergOPT`based fine-tuning improves performance over standard fine-tuning by 2.34%, 5.51%, 4.45%, and 1.56% under Weight Averaging, Task Arithmetic, TIES-Merging, and DARE, respectively. These results demonstrate that our method exhibits robust cross-architecture generalization.

### C.2 APPLICATIONS IN LARGE-SCALE LLM ARCHITECTURE

In the main text, we primarily conduct experiments on Llama-1B/3B and Qwen2.5-1.5B. To further validate the effectiveness of our approach on larger-scale architectures, in this section we additionally present results on Llama-8B. Specifically, we merge expert models fine-tuned on C-STANCE, FOMC, ScienceQA, and NumGLUE-cm, and apply our proposed method to four representative model

Table 7: Performance comparison of model merging methods with Qwen2.5-1.5B-Instruct.

| Method | Task Performance | | | | | | | Avg. (↑) |
|---|---|---|---|---|---|---|---|---|
| | C-STANCE | FOMC | MeetingBank | ScienceQA | NumGLUE-cm | NumGLUE-ds | 20Minuten | |
| Pre-Trained | 0.4700 | 0.3569 | 0.1895 | 0.7182 | 0.0976 | 0.2195 | 0.3885 | 0.3486 |
| Standard Fine-Tuned | 0.5385 | 0.6230 | 0.3190 | 0.8439 | 0.3902 | 0.4695 | 0.3956 | 0.4974 |
| `MergOPT` Fine-Tuned | 0.5250 | 0.6351 | 0.3459 | 0.8775 | 0.5122 | 0.4390 | 0.3977 | 0.5336 |
| Weight Averaging | 0.4880 | 0.4012 | 0.1967 | 0.7500 | 0.4390 | 0.3841 | 0.3920 | 0.4359 |
| **Weight Averaging w/ `MergOPT`** | 0.5005 | 0.4536 | 0.1994 | 0.7530 | 0.4390 | 0.3841 | 0.3928 | 0.4461(+2.34%) |
| Task Arithmetic | 0.5055 | 0.4677 | 0.2189 | 0.7540 | 0.4146 | 0.3902 | 0.3880 | 0.4484 |
| **Task Arithmetic w/ `MergOPT`** | 0.5220 | 0.5121 | 0.2205 | 0.7625 | 0.5122 | 0.3963 | 0.3859 | 0.4731(+5.51%) |
| TIES-Merging | 0.5160 | 0.4597 | 0.2253 | 0.7475 | 0.4390 | 0.4024 | 0.3875 | 0.4539 |
| **TIES-Merging w/ `MergOPT`** | 0.5260 | 0.5081 | 0.2210 | 0.7660 | 0.5122 | 0.3963 | 0.3890 | 0.4741(+4.45%) |
| DARE | 0.5035 | 0.4536 | 0.2260 | 0.7425 | 0.4390 | 0.3963 | 0.3829 | 0.4491 |
| **DARE w/ `MergOPT`** | 0.5100 | 0.5161 | 0.2174 | 0.7710 | 0.4390 | 0.3598 | 0.3794 | 0.4561(+1.56%) |

merging techniques: Weight Averaging, Task Arithmetic, TIES-Merging, and DARE. As shown in Table 8, `MergOPT` yields average performance improvements of 1.08%, 1.40%, 0.46%, and 0.51% for Weight Averaging, Task Arithmetic, TIES-Merging, and DARE, respectively. These results provide strong evidence that our method remains effective at larger parameter scales.

Table 8: Performance Comparison of Model Merging Methods on Llama3.1-8B-Instruct.

| Method | Task Performance | | | | Avg. (↑) |
|---|---|---|---|---|---|
| | C-STANCE | FOMC | ScienceQA | NumGLUE-cm | |
| Pre-Trained | 0.4197 | 0.3085 | 0.9075 | 0.2927 | 0.4821 |
| Standard Fine-Tuned | 0.5712 | 0.7234 | 0.9418 | 0.6721 | 0.7271 |
| `MergOPT` Fine-Tuned | 0.5637 | 0.7335 | 0.9403 | 0.6234 | 0.7152 |
| Weight Averaging | 0.4982 | 0.6489 | 0.9293 | 0.7319 | 0.7021 |
| **Weight Averaging w/ `MergOPT`** | 0.5071 | 0.6627 | 0.9307 | 0.7384 | 0.7097(+1.08%) |
| Task Arithmetic | 0.5268 | 0.6731 | 0.9218 | 0.6833 | 0.7012 |
| **Task Arithmetic w/ `MergOPT`** | 0.5318 | 0.6803 | 0.9251 | 0.7069 | 0.7110(+1.40%) |
| TIES-Merging | 0.5193 | 0.6792 | 0.9212 | 0.7074 | 0.7068 |
| **TIES-Merging w/ `MergOPT`** | 0.5236 | 0.6824 | 0.9237 | 0.7106 | 0.7101(+0.46%) |
| DARE | 0.5227 | 0.6696 | 0.9168 | 0.7071 | 0.7041 |
| **DARE w/ `MergOPT`** | 0.5259 | 0.6765 | 0.9192 | 0.7092 | 0.7077(+0.51%) |

## C.3 APPLICATIONS IN VISUAL ARCHITECTURE AND TASKS

Beyond language tasks, vision tasks are also a major application area for model merging methods. To evaluate the effectiveness of our approach across different domains, we further conduct experiments in the vision setting.

**Architecture and Datasets.** We follow standard configurations used in prior work on vision model merging (Ilharco et al., 2023), adopting CLIP-ViT-B/32 (Dosovitskiy et al., 2021; Radford et al., 2021) as the base model and fine-tuning it into expert models on ten downstream tasks: SUN397 (Xiao et al., 2016), Cars (Krause et al., 2013), RESISC45 (Cheng et al., 2017), EuroSAT (Helber et al., 2019), SVHN (Yuval, 2011), GTSRB (Stallkamp et al., 2011), MNIST (LeCun, 1998), DTD (Cimpoi et al., 2014), Flowers102 (Nilsback & Zisserman, 2008), and PCAM (Veeling et al., 2018). As all tasks are classification tasks, we adopt Top-1 classification accuracy as the unified evaluation metric and report the average value across all tasks.

**Fine-tuning Methods.** We use AdamW as the standard fine-tuning baseline, with hyperparameters following previous work. In addition, we consider fine-tuning in the tangent space (Ortiz-Jimenez et al., 2023) as well as our proposed `MergOPT`-based fine-tuning scheme.

**Merging Methods.** Consistent with our LLM experiments, we compare four representative model merging approaches: Weight Averaging (Wortsman et al., 2022), Task Arithmetic (Ilharco et al., 2023), TIES-Merging (Yadav et al., 2023), and DARE (Yu et al., 2024). Moreover, we include WUDI-Merging (Cheng et al., 2025), a recent optimization-based, data-free merging method that has been shown to significantly outperform several typical model merging baselines.

**Performance Comparison.** As shown in Table 9, we make the following observations: (i) Compared with standard fine-tuning, the proposed `MergOPT` significantly improves performance when merging 10 tasks. For example, under five representative merging methods—Weight Averaging, Task Arithmetic, TIES-Merging, DARE, and WUDI-Merging—`MergOPT` yields average gains of 2.36%, 3.43%, 4.10%, 4.05%, and 1.32%, respectively. (ii) Fine-tuning in the tangent space allows a better decoupling between the input space and the weight space, thereby reducing interference during merging (Ortiz-Jimenez et al., 2023)[1]. We observe that, in most cases, tangent-space fine-tuning indeed improves performance over standard fine-tuning: under Task Arithmetic, TIES-Merging, and DARE, it brings gains of 3.13%, 3.95%, and 3.00%, respectively, although these improvements are still lower than those achieved by MergOPT in the corresponding settings. However, under Weight Averaging and WUDI-Merging, the merged models exhibit some performance degradation; we leave a deeper investigation of this phenomenon to future work. (iii) WUDI-Merging shows consistently stronger performance than other merging baselines, as it constructs a data-free optimization objective and explicitly optimizes the merged model parameters. Overall, these results provide consistent evidence that our proposed method is broadly applicable across diverse domains.

Table 9: Performance Comparison of Model Merging Methods on Visual Tasks with CLIP-ViT-B/32.

| Method | SUN397 | Cars | RESISC45 | EuroSAT | SVHN | GTSRB | MNIST | DTD | Flowers102 | PCAM | Avg. (↑) |
|---|---|---|---|---|---|---|---|---|---|---|---|
| Weight Averaging (Standard) | 60.7 | 54.4 | 59.0 | 35.9 | 33.4 | 33.2 | 58.7 | 42.0 | 77.5 | 94.3 | 54.9 |
| Weight Averaging (Tangent) | 56.1 | 51.2 | 53.7 | 36.4 | 26.3 | 29.8 | 58.2 | 44.1 | 68.6 | 80.3 | 50.5 (-8.01%) |
| **Weight Averaging (`MergOPT`)** | 60.8 | 56.6 | 59.7 | 36.9 | 32.9 | 35.5 | 66.9 | 42.6 | 77.5 | 92.3 | 56.2 (+2.36%) |
| Task Arithmetic (Standard) | 62.1 | 57.0 | 72.0 | 77.7 | 64.5 | 59.0 | 91.4 | 46.3 | 68.6 | 70.5 | 66.9 |
| Task Arithmetic (Tangent) | 62.7 | 64.1 | 77.8 | 89.3 | 54.7 | 55.9 | 84.5 | 52.7 | 72.5 | 76.0 | 69.0 (+3.13%) |
| **Task Arithmetic (`MergOPT`)** | 57.0 | 55.5 | 70.0 | 71.8 | 76.6 | 74.6 | 95.7 | 47.3 | 63.7 | 80.0 | 69.2 (+3.43%) |
| TIES-Merging (Standard) | 64.6 | 64.9 | 70.6 | 74.7 | 62.8 | 55.4 | 92.0 | 44.1 | 65.7 | 62.8 | 65.8 |
| TIES-Merging (Tangent) | 63.8 | 65.0 | 78.1 | 90.1 | 54.0 | 55.9 | 85.0 | 50.5 | 70.6 | 71.5 | 68.4 (+3.95%) |
| **TIES-Merging (`MergOPT`)** | 56.2 | 58.1 | 70.0 | 66.7 | 78.0 | 73.0 | 96.7 | 45.2 | 64.7 | 75.8 | 68.5 (+4.10%) |
| DARE (Standard) | 61.1 | 56.9 | 71.5 | 78.5 | 65.6 | 56.7 | 92.1 | 45.2 | 67.6 | 71.6 | 66.7 |
| DARE (Tangent) | 62.7 | 64.5 | 77.1 | 90.1 | 51.3 | 56.7 | 85.2 | 51.6 | 72.5 | 76.1 | 68.7 (+3.00%) |
| **DARE (`MergOPT`)** | 59.3 | 55.4 | 70.2 | 74.2 | 74.8 | 73.2 | 95.1 | 48.4 | 64.7 | 78.4 | 69.4 (+4.05%) |
| WUDI-Merging (Standard) | 67.5 | 72.6 | 84.0 | 93.8 | 89.4 | 95.8 | 99.2 | 61.2 | 77.5 | 91.2 | 83.2 |
| WUDI-Merging (Tangent) | 66.8 | 67.0 | 81.9 | 92.1 | 76.3 | 79.7 | 94.4 | 61.2 | 80.4 | 71.2 | 77.1 (-7.33%) |
| **WUDI-Merging (`MergOPT`)** | 70.2 | 69.9 | 86.6 | 92.9 | 90.7 | 97.3 | 99.0 | 63.8 | 79.4 | 93.0 | 84.3 (+1.32%) |

# D  ADDITIONAL EXPERIMENTAL ANALYSIS

In this section, we provide a comprehensive analysis of the feasible region of hyperparameters in our proposed `MergOPT` method (Sec. D.1) and compare its performance with different optimizers (Sec. D.2). Next, we present additional experimental results on various task combinations (Sec. D.3). Finally, in Appendix D.4, we discussed the sensitivity of the proposed method to the hyperparameters.

## D.1  ANALYSIS OF THE FEASIBLE REGION OF HYPERPARAMETERS

### D.1.1  ANALYSIS ON TASK VECTORS $\Delta\theta$

Our `MergOPT` approach requires treating task vectors from other expert models as merge-induced parameter shifts during single-task fine-tuning, in order to simulate potential merging disturbances. However, in practice, such task vectors from other experts are typically inaccessible. To address this, we conduct a detailed analysis of the distributional properties of task vectors. Interestingly, we find that these vectors can be well approximated by a Laplace distribution $Laplace(\mu, b)$, where $\mu$ denotes the location parameter corresponding to the mean of the task vectors, and $b$ represents the scale parameter that characterizes their dispersion around the mean.

To validate this, we visualize the distribution of task vectors $\{\Delta\theta_1, \Delta\theta_2, \ldots, \Delta\theta_K\}$[2] obtained from fine-tuning Llama-3.2-1B-Instruct on the six (i.e., ScienceQA, FOMC, MeetingBank, C-STANCE,

---

[1]It is important to note that the inference cost of linear models obtained via tangent-space fine-tuning is typically $2 - 3\times$ higher than that of standard fine-tuning. For details, please refer to the Computational complexity section in Appendix B of the original paper (Ortiz-Jimenez et al., 2023). In contrast, our method does not incur additional inference costs for the model.

[2]Since the dimensionality of task vectors matches that of model parameters, direct visualization of the complete task vector distribution is infeasible. To address this, we randomly sampled one million parameters from the entire task vector for visualization purposes.

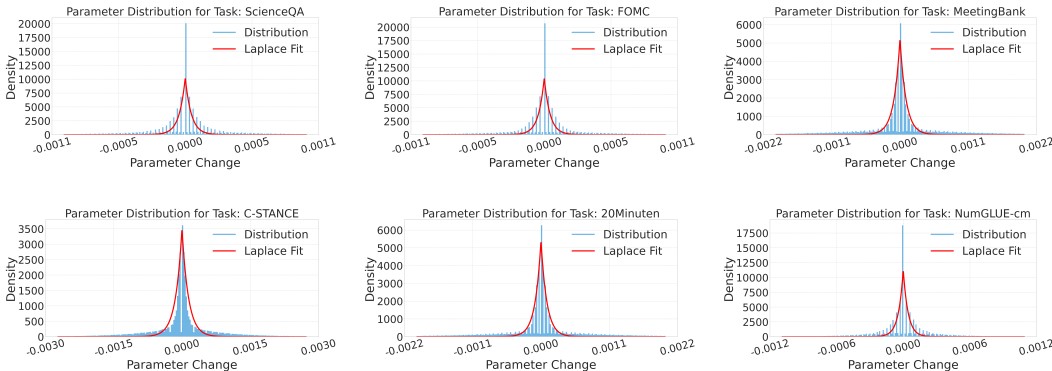

Figure 3: Distribution of task vectors for different tasks. The histograms show the empirical distributions of task vectors (blue), along with the fitted Laplace distributions (red).

20Minuten, and NumGLUE-cm) tasks, as shown in Figure 3. The histograms illustrate that the empirical distributions (blue color) of these task vectors closely align[3] with the fitted Laplace distributions (red color), confirming our assumption. Based on this observation, we set the location parameter $\mu$ to 0 (the mean of the task vectors) and treat the scale parameter $b$ as a tunable hyperparameter in our method. Table 11 further explores the impact of different $b$ values on merging performance.

Moreover, as shown in Figure 1, we also observe that the accumulated task vectors (i.e., $\sum_{i=1}^{K} \Delta\theta_i$) closely follow a Laplace distribution. This finding holds consistently across three mainstream architectures, namely Llama-3.2-1B-Instruct, Llama-3.2-3B-Instruct, and Qwen2.5-1.5B-Instruct. This property enables us to approximate real merging scenarios during expert fine-tuning by sampling from the corresponding Laplace distribution, thereby effectively simulating the parameter changes induced by accumulated task vectors.

The above analysis is primarily based on visual inspection and empirical fitting of the task-vector distributions. To provide a more rigorous statistical validation, we additionally randomly sample three tasks and, for each, compute the Kolmogorov–Smirnov (K–S) distance and the average log-likelihood to quantify the goodness-of-fit of the Laplace distribution. As shown in Table 10, the K–S distances for the three tasks are 0.101, 0.106, and 0.106, all around 0.10, indicating a reasonably close match between the Laplace distribution and the empirical distributions in terms of overall shape. Meanwhile, the corresponding average log-likelihoods are 5.76, 5.87, and 6.54, which are relatively high and stable, further suggesting that the Laplace distribution provides a good approximation to the main mass of the task-vector distributions.

It is important to emphasize that MergOPT does not rely on the Laplace distribution to perfectly capture the full, true distribution of task vectors. Instead, Laplace is adopted as a structurally simple and easily sampled approximation to potential worst-case merge offsets, enabling a distributionally robust optimization objective in weight space.

Table 10: K–S Distance and Average Log Likelihood for Each Task Vector.

| Task | K-S Distance | Avg. Log Likelihood |
|------|--------------|---------------------|
| C-STANCE | 0.101 | 5.76 |
| MeetingBank | 0.106 | 5.87 |
| ScienceQA | 0.106 | 6.54 |

---

[3]It is worth emphasizing that the empirical distribution in the figure does not perfectly fit a Laplace distribution; there are noticeable deviations, especially a slight underestimation of the tail mass. In practice, optimization in deep learning is highly complex, and the distribution of parameter updates induced by optimization is difficult to capture exactly with any simple explicit distribution.

### D.1.2    ANALYSIS ON SCALE PARAMETER $b$ OF LAPLACE DISTRIBUTION

Table 11 presents the performance of Task Arithmetic merging on seven models using Llama-3.2-1B-Instruct, under different values of the Laplace scale parameter $b$. We observe that our proposed `MergOPT` method consistently enhances merging performance across a range of $b$ values, demonstrating its robustness to this hyperparameter. Notably, setting $b = 0.0005$ yields the best average score of 0.4164, representing a +2.68% relative improvement over the task arithmetic baseline (average score of 0.4055). This indicates that our approach effectively simulates merging shifts during fine-tuning, leading to more robust merged models.

### D.1.3    ANALYSIS ON LOCATION PARAMETER $\mu$ OF LAPLACE DISTRIBUTION

Based on the results in Figures 1 and 3, we empirically observe that the task vectors approximately follow a $\text{Laplace}(\mu, b)$ distribution with $\mu$ concentrated around 0. Therefore, in our main experiments, we set $\mu = 0$ by default. In this section, we further analyze the impact of shifting the mean $\mu$ in the Laplace distribution on performance. As shown in Table 12, under both Task Arithmetic and TIES-Merging, performance consistently degrades when $\mu$ deviates from 0. For example, when merging two models with Task Arithmetic, the average scores with $\mu = 0.1$ and $\mu = -0.1$ are 0.4878 and 0.4877, both lower than 0.4902 obtained with $\mu = 0$. Similarly, for TIES-Merging, the scores with $\mu = 0.1$ and $\mu = -0.1$ are 0.4910 and 0.4912, compared to 0.4929 when $\mu = 0$. These results indicate that setting $\mu = 0$ is a well-justified default choice that better matches the empirical distribution of task vectors.

Table 11: Task Arithmetic Merging Results on Llama-3.2-1B-Instruct with Varying Laplace Scale $b$.

| $b$ | Task Performance | | | | | | | Avg. ($\uparrow$) |
| --- | --- | --- | --- | --- | --- | --- | --- | --- |
| | C-STANCE | FOMC | MeetingBank | ScienceQA | NumGLUE-cm | NumGLUE-ds | 20Minuten | |
| (base) | 0.4219 | 0.4980 | 0.2094 | 0.7370 | 0.2439 | 0.3476 | 0.3805 | 0.4055 |
| 0.0005 (Default) | 0.3995 | 0.5101 | 0.2039 | 0.7499 | 0.2683 | 0.4024 | 0.3808 | 0.4164 |
| 0.001 | 0.3991 | 0.5040 | 0.2060 | 0.7479 | 0.2439 | 0.4024 | 0.3801 | 0.4119 |
| 0.05 | 0.3991 | 0.5020 | 0.2014 | 0.7365 | 0.2683 | 0.4024 | 0.3861 | 0.4137 |

Table 12: Task Arithmetic Merging Results on Llama-3.2-1B-Instruct with Varying Location $\mu$

| Method | $\mu = -0.1$ | | | $\mu = 0$ (Default) | | | $\mu = 0.1$ | | |
| --- | --- | --- | --- | --- | --- | --- | --- | --- | --- |
| | C-STANCE | FOMC | Avg. ($\uparrow$) | C-STANCE | FOMC | Avg. ($\uparrow$) | C-STANCE | FOMC | Avg. ($\uparrow$) |
| **Task Arithmetic w/ `MergOPT`** | 0.4417 | 0.5339 | 0.4878 | 0.4440 | 0.5363 | 0.4902 | 0.4423 | 0.5331 | 0.4877 |
| **TIES-Merging w/ `MergOPT`** | 0.4396 | 0.5423 | 0.4910 | 0.4414 | 0.5444 | 0.4929 | 0.4408 | 0.5416 | 0.4912 |

### D.1.4    ANALYSIS ON MERGING COEFFICIENT $\alpha$

Table 13 reports the performance of Task Arithmetic merging on the Llama-3.2-1B-Instruct model under different merging coefficients $\alpha$. We observe that smaller merging coefficients (i.e., $\alpha$ in Eq. 2) generally yield better results. For example, when $\alpha$ ranges from 0.1 to 0.6, the merged models consistently achieve scores above 0.34, with the best performance of 0.4055 attained at $\alpha = 0.2$. In contrast, larger coefficients cause severe performance degradation, with results even falling below those of the pretrained model (e.g., 0.3064). In particular, when $\alpha = 1.0$, the merged model drops to 0.2194, which is approximately half of the best score. These findings suggest that optimal merging coefficients are typically small. Accordingly, we adopt the range $[0.1, 0.6]$ as the feasible set (i.e., $\mathcal{A}$) of coefficients throughout this work.

### D.2    COMPARISON WITH DIFFERENT OPTIMIZERS

### D.2.1    COMPARISON WITH SAM OPTIMIZER

SAFT-Merge (Lee et al., 2025) is a robust model merging approach that employs the Sharpness-Aware Minimization (SAM) (Foret et al., 2021) or Adaptive SAM (ASAM) (Kwon et al., 2021) optimizer during fine-tuning to seek flatter minima, thereby reducing performance degradation at the merging

Table 13: Task Arithmetic Merging Results on Llama-3.2-1B-Instruct with Varying Merging Coefficient.

| $\alpha$ | Task Performance | | | | | | | Avg. ($\uparrow$) |
|---|---|---|---|---|---|---|---|---|
| | C-STANCE | FOMC | MeetingBank | ScienceQA | NumGLUE-cm | NumGLUE-ds | 20Minuten | |
| Pre-Trained | 0.3386 | 0.2581 | 0.2036 | 0.6780 | 0.1220 | 0.1646 | 0.3802 | 0.3064 |
| 0.1 | 0.4098 | 0.4456 | 0.2094 | 0.7125 | 0.1951 | 0.2927 | 0.3878 | 0.3790 |
| 0.2 | **0.4220** | **0.4980** | 0.2094 | **0.7370** | **0.2439** | 0.3476 | 0.3805 | **0.4055** |
| 0.3 | 0.4120 | 0.4718 | 0.2274 | 0.7355 | 0.1951 | 0.3902 | **0.3809** | 0.4019 |
| 0.4 | 0.4104 | 0.4355 | **0.2336** | 0.7170 | **0.2439** | **0.3963** | 0.3772 | 0.4020 |
| 0.5 | 0.3931 | 0.4476 | 0.2242 | 0.6735 | 0.1707 | 0.3902 | 0.3790 | 0.3826 |
| 0.6 | 0.3517 | 0.4395 | 0.1758 | 0.6264 | 0.0976 | 0.3598 | 0.3783 | 0.3470 |
| 0.7 | 0.3503 | 0.3952 | 0.1231 | 0.5615 | 0.0976 | 0.3354 | 0.3718 | 0.3193 |
| 0.8 | 0.3090 | 0.3690 | 0.0949 | 0.4667 | 0.0000 | 0.3171 | 0.3687 | 0.2750 |
| 0.9 | 0.2998 | 0.2621 | 0.0895 | 0.4094 | 0.0488 | 0.2500 | 0.3665 | 0.2466 |
| 1.0 | 0.3138 | 0.2823 | 0.0848 | 0.3611 | 0.0000 | 0.1280 | 0.3658 | 0.2194 |

stage. In this section, we compare our proposed `MergOPT` with SAFT-Merge from two perspectives: equal training epochs and comparable fine-tuning time.

**Equal Training Epochs.** As shown in Table 14, when all optimizers are trained for the same number of parameter updates, the results exhibit the following trends: Under Weight Averaging and DARE, our method surpasses SAFT-Merge (SAM) by 2.59% and 5.20%, respectively. In contrast, SAFT-Merge (SAM) slightly outperforms `MergOPT` under Task Arithmetic and TIES-Merging, with relative gains of 2.24% and 1.29%. Furthermore, when comparing SAFT-Merge (SAM) and SAFT-Merge (ASAM), we observe that ASAM is generally more effective: under Weight Averaging, TIES-Merging, and DARE, it achieves improvements of 1.68%, 1.17%, and 4.03%, respectively. This advantage may stem from the adaptive perturbation radius used in ASAM, as opposed to that in SAM. However, as shown in Table 15, SAFT-Merge (SAM) and SAFT-Merge (ASAM) require an average runtime that is 2.04× and 2.06× AdamW, while `MergOPT` only incurs 1.17× AdamW overhead. SAFT-Merge (SAM) and SAFT-Merge (ASAM) are computationally expensive because they require one gradient ascent step to compute the perturbation direction at each parameter update step, followed by one gradient descent step for parameter updates. Consequently, each update step involves two full forward-backward propagation passes, leading to a computational cost roughly twice that of standard fine-tuning. In contrast, our `MergOPT` method directly samples the perturbation direction, resulting in a computational cost much closer to that of standard fine-tuning. This highlights that `MergOPT` offers better efficiency while still delivering competitive robustness.

**Comparable Fine-tuning Time.** To ensure a fair comparison, we further align the computational cost of SAM (ASAM) and `MergOPT` (1.04× and 1.17× AdamW, respectively, see Table 17). As shown in Table 16, in this setting, the performance improvements become more consistent: With Weight Averaging, Task Arithmetic, TIES-Merging, and DARE, `MergOPT` achieves relative gains of 4.08%, 0.70%, 0.95%, and 3.98% over SAFT-Merge (SAM), respectively. Compared with SAFT-Merge (SAM), SAFT-Merge (ASAM) shows only limited gains under Weight Averaging, Task Arithmetic, and TIES-Merging. This may be because, when restricted to a training budget comparable to AdamW, the optimizations performed by ASAM and SAM are less thorough than those of AdamW, which in turn limits their performance. These results demonstrate that at comparable computational cost, `MergOPT` consistently outperforms SAFT-Merge (SAM) and SAFT-Merge (ASAM), striking a better balance between efficiency and robustness.

### D.2.2 COMBINED WITH OTHER BASE OPTIMIZER

While AdamW is our default base optimizer, `MergOPT` is optimizer-agnostic and can be combined with any standard optimizer. To assess this generality, we also instantiate `MergOPT` with SGD. Table 18 reports results on Llama-3.2-1B-Instruct when SGD is used as the base optimizer.

Relative to the naive SGD fine-tuning, `MergOPT` yields more robust post-merge performance for most merging procedures: Task Arithmetic improves from 0.3286 to 0.3566 (+8.52%), TIES-Merging from 0.3225 to 0.3552 (+10.14%), and DARE from 0.2313 to 0.2471 (+6.83%), while Weight Averaging decreases slightly from 0.3634 to 0.3468 (–4.56%). We also note that both `MergOPT` and standard SGD achieve relatively low absolute scores under DARE (0.2471 vs. 0.2313). A plausible

Table 14: Performance comparison of `MergOPT` and SAFT-Merge (based on SAM and ASAM) under equal training epochs on Llama-3.2-1B-Instruct. Notably, SAM incurs $2.04\times$ the optimization cost of standard AdamW, whereas ours incurs only $1.17\times$.

| Method | Task Performance | | | | | | | Avg. ($\uparrow$) |
|---|---|---|---|---|---|---|---|---|
| | C-STANCE | FOMC | MeetingBank | ScienceQA | NumGLUE-cm | NumGLUE-ds | 20Minuten | |
| Fine-Tuned w/ SAFT-Merge (SAM) | 0.4820 | 0.5927 | 0.3354 | 0.8724 | 0.5122 | 0.5488 | 0.3838 | 0.5325 |
| Fine-Tuned w/ SAFT-Merge (ASAM) | 0.4867 | 0.6005 | 0.3287 | 0.8651 | 0.4878 | 0.5215 | 0.3814 | 0.5245 |
| Fine-Tuned w/ `MergOPT` | 0.4957 | 0.6331 | 0.3158 | 0.8780 | 0.4390 | 0.5305 | 0.3829 | 0.5250 |
| Weight Averaging w/ SAFT-Merge (SAM) | 0.4320 | 0.4355 | 0.2108 | 0.7075 | 0.2927 | 0.3293 | 0.3781 | 0.3980 |
| Weight Averaging w/ SAFT-Merge (ASAM) | 0.4362 | 0.4487 | 0.2138 | 0.7143 | 0.3048 | 0.3356 | 0.3796 | 0.4047$_{(+1.68\%)}$ |
| **Weight Averaging w/ `MergOPT`** | 0.4130 | 0.4819 | 0.2153 | 0.7530 | 0.2927 | 0.3171 | 0.3850 | 0.4083$_{(+2.59\%)}$ |
| Task Arithmetic w/ SAFT-Merge (SAM) | 0.4365 | 0.4839 | 0.2031 | 0.7100 | 0.3902 | 0.3720 | 0.3861 | 0.4260 |
| Task Arithmetic w/ SAFT-Merge (ASAM) | 0.4280 | 0.4798 | 0.2027 | 0.6885 | 0.3415 | 0.3598 | 0.3826 | 0.4119$_{(-3.30\%)}$ |
| **Task Arithmetic w/ `MergOPT`** | 0.4203 | 0.4718 | 0.2077 | 0.7530 | 0.3171 | 0.3659 | 0.3797 | 0.4165$_{(-2.24\%)}$ |
| TIES-Merging w/ SAFT-Merge (SAM) | 0.4395 | 0.4698 | 0.2063 | 0.7232 | 0.3415 | 0.3598 | 0.3836 | 0.4177 |
| TIES-Merging w/ SAFT-Merge (ASAM) | 0.4423 | 0.4752 | 0.2089 | 0.7268 | 0.3561 | 0.3648 | 0.3843 | 0.4226$_{(+1.17\%)}$ |
| **TIES-Merging w/ `MergOPT`** | 0.4202 | 0.4536 | 0.2093 | 0.7555 | 0.2927 | 0.3720 | 0.3825 | 0.4123$_{(-1.29\%)}$ |
| DARE w/ SAFT-Merge (SAM) | 0.4240 | 0.4234 | 0.2079 | 0.6895 | 0.2927 | 0.3415 | 0.3807 | 0.3942 |
| DARE w/ SAFT-Merge (ASAM) | 0.4318 | 0.4615 | 0.2037 | 0.7142 | 0.3268 | 0.3512 | 0.3815 | 0.4101$_{(+4.03\%)}$ |
| **DARE w/ `MergOPT`** | 0.4192 | 0.4819 | 0.2107 | 0.7485 | 0.2927 | 0.3659 | 0.3843 | 0.4147$_{(+5.20\%)}$ |

Table 15: Average fine-tuning time (seconds) of AdamW, SAFT-Merge (based on SAM and ASAM), and `MergOPT` under equal training epochs.

| Optimizer | Task Fine-tuning Time (s) | | | | | | | Avg. ($\downarrow$) |
|---|---|---|---|---|---|---|---|---|
| | C-STANCE | FOMC | MeetingBank | ScienceQA | NumGLUE-cm | NumGLUE-ds | 20Minuten | |
| AdamW | 1927.74 | 1155.33 | 2688.37 | 1156.57 | 1923.30 | 1921.97 | 2684.57 | 1922.55 |
| SAFT-Merge (SAM) | 3933.49 | 2360.41 | 5497.92 | 2363.90 | 3927.65 | 3927.04 | 5488.63 | 3928.43$_{(2.04\times)}$ |
| SAFT-Merge (ASAM) | 3968.43 | 2377.58 | 5531.46 | 2378.92 | 3958.27 | 3953.68 | 5527.37 | 3956.54$_{(2.06\times)}$ |
| **`MergOPT`** | 2247.64 | 1348.34 | 3147.20 | 1349.36 | 2241.04 | 2240.76 | 3141.03 | 2245.05$_{(1.17\times)}$ |

explanation is that DARE's stochastic masking of task-vector coordinates can inadvertently suppress salient parameters, leading to nontrivial information loss during merging.

### D.2.3 COMPARATIVE ANALYSIS OF MODEL MERGING VIA DIFFERENT OPTIMIZATION METHODS

In our main experiments, we assume that all models are fine-tuned either with standard AdamW or with the proposed MergOPT. A natural question is: *can models fine-tuned with standard AdamW and with MergOPT be merged together effectively*? To investigate this, we conduct experiments on Llama3.2-1B-Instruct. Specifically, we randomly select two tasks, NumGLUE-cm and NumGLUE-ds, and fine-tune models on each task using both AdamW and MergOPT. We then evaluate the following three merging settings: (i) AdamW+AdamW: merging two models fine-tuned with standard AdamW; (ii) AdamW+MergOPT: merging one AdamW fine-tuned model with one MergOPT fine-tuned model; and (iii) MergOPT+MergOPT: merging two models fine-tuned with MergOPT.

As shown in Table 19, we observe that even when only one of the two models (AdamW+MergOPT) is fine-tuned with MergOPT, the merged model already achieves noticeably better performance than merging two standard AdamW models (AdamW+AdamW). For example, under Task Arithmetic, AdamW+MergOPT attains an average score of 0.4116, compared to 0.3994 for AdamW+AdamW. When both models are fine-tuned with MergOPT, the performance further improves. These results indicate that MergOPT is compatible with and beneficial for merging models obtained from heterogeneous fine-tuning strategies.

### D.3 ABLATION STUDY ON NUMBER OF TASKS TO MERGE

In Table 3 of the main text, we have already reported the results for the 4-task merging scenario. To further examine the effectiveness of our approach under different task scales, this section extends the study to 2-task and 6-task merging. Specifically, we randomly sampled subsets from the full set of seven tasks, with each subset containing either 2 or 6 tasks. We then merged the expert models fine-tuned with the standard AdamW optimizer, as well as those fine-tuned with our proposed `MergOPT`, and evaluated their performance.

The results are presented in Tables 20 and 21. Overall, we observe the following: (i) 2-task merging, as shown in Table 20: For both Task Arithmetic and TIES-Merging, incorporating `MergOPT`

Table 16: Performance comparison of `MergOPT` and SAFT-Merge (based on SAM and ASAM) under comparable fine-tuning cost on Llama-3.2-1B-Instruct.

| Method | Task Performance | | | | | | | Avg. (↑) |
|---|---|---|---|---|---|---|---|---|
| | C-STANCE | FOMC | MeetingBank | ScienceQA | NumGLUE-cm | NumGLUE-ds | 20Minuten | |
| Fine-Tuned w/ SAFT-Merge (SAM) | 0.4940 | 0.5746 | 0.3616 | 0.8305 | 0.4390 | 0.5671 | 0.3850 | 0.5217 |
| Fine-Tuned w/ SAFT-Merge (ASAM) | 0.4829 | 0.5874 | 0.3725 | 0.8452 | 0.4634 | 0.5842 | 0.3897 | 0.5322 |
| `MergOPT` Fine-Tuned | 0.4957 | 0.6331 | 0.3158 | 0.8780 | 0.4390 | 0.5305 | 0.3829 | 0.5250 |
| Weight Averaging w/ SAFT-Merge (SAM) | 0.4245 | 0.4315 | 0.2149 | 0.6930 | 0.2683 | 0.3293 | 0.3846 | 0.3923 |
| Weight Averaging w/ SAFT-Merge (ASAM) | 0.4198 | 0.4267 | 0.2127 | 0.6852 | 0.2631 | 0.3214 | 0.3821 | 0.3873$_{(-1.27\%)}$ |
| **Weight Averaging w/ `MergOPT`** | 0.4130 | 0.4819 | 0.2153 | 0.7530 | 0.2927 | 0.3171 | 0.3850 | 0.4083$_{(+4.08\%)}$ |
| Task Arithmetic SAFT-Merge (SAM) | 0.4285 | 0.4919 | 0.2075 | 0.6840 | 0.3415 | 0.3598 | 0.3822 | 0.4136 |
| Task Arithmetic w/ SAFT-Merge (ASAM) | 0.4316 | 0.4967 | 0.2098 | 0.6942 | 0.3456 | 0.3651 | 0.3837 | 0.4181$_{(+1.08\%)}$ |
| **Task Arithmetic w/ `MergOPT`** | 0.4203 | 0.4718 | 0.2077 | 0.7530 | 0.3171 | 0.3659 | 0.3797 | 0.4165$_{(+0.70\%)}$ |
| TIES-Merging w/ SAFT-Merge (SAM) | 0.4295 | 0.4758 | 0.2075 | 0.6930 | 0.3171 | 0.3537 | 0.3826 | 0.4084 |
| TIES-Merging w/ SAFT-Merge (ASAM) | 0.4251 | 0.4698 | 0.2043 | 0.6837 | 0.3109 | 0.3472 | 0.3798 | 0.4030$_{(-1.32\%)}$ |
| **TIES-Merging w/ `MergOPT`** | 0.4202 | 0.4536 | 0.2093 | 0.7555 | 0.2927 | 0.3720 | 0.3825 | 0.4123$_{(+0.95\%)}$ |
| DARE w/ SAFT-Merge (SAM) | 0.4335 | 0.4395 | 0.2134 | 0.6630 | 0.3171 | 0.3476 | 0.3776 | 0.3988 |
| DARE w/ SAFT-Merge (ASAM) | 0.4298 | 0.4351 | 0.2108 | 0.6548 | 0.3127 | 0.3421 | 0.3751 | 0.3943$_{(-1.12\%)}$ |
| **DARE w/ `MergOPT`** | 0.4192 | 0.4819 | 0.2107 | 0.7485 | 0.2927 | 0.3659 | 0.3843 | 0.4147$_{(+3.98\%)}$ |

Table 17: Average fine-tuning time (seconds) of AdamW, SAFT-Merge (based on SAM and ASAM), and `MergOPT` under comparable fine-tuning cost settings.

| Optimizer | Task Fine-tuning Time (s) | | | | | | | Avg. (↓) |
|---|---|---|---|---|---|---|---|---|
| | C-STANCE | FOMC | MeetingBank | ScienceQA | NumGLUE-cm | NumGLUE-ds | 20Minuten | |
| AdamW | 1927.74 | 1155.33 | 2688.37 | 1156.57 | 1923.30 | 1921.97 | 2684.57 | 1922.55 |
| SAFT-Merge (SAM) | 1994.25 | 1196.54 | 2788.16 | 1197.83 | 1990.56 | 1994.05 | 2782.05 | 1991.92$_{(1.04\times)}$ |
| SAFT-Merge (ASAM) | 1923.38 | 1153.39 | 2782.19 | 1254.05 | 2018.30 | 2020.72 | 2782.95 | 1990.71$_{(1.04\times)}$ |
| **`MergOPT`** | 2247.64 | 1348.34 | 3147.20 | 1349.36 | 2241.04 | 2240.76 | 3141.03 | 2245.05$_{(1.17\times)}$ |

consistently improves the average performance of the merged models, with the maximum gain exceeding 8%. (ii) 6-task merging, as shown in Table 21: Compared to standard fine-tuning, `MergOPT` also provides stable improvements across different task groups, with relative performance gains ranging from 2.57% to 6.78%. These findings indicate that `MergOPT` remains robust across both small- and large-scale merging scenarios, further validating its generality and reliability.

## D.4 HYPERPARAMETER SENSITIVITY ANALYSIS

### D.4.1 DIFFERENT TASK VECTOR DISTRIBUTIONS

In this work, our analysis suggests that the Laplace distribution provides a better fit to the empirical task-vector distribution. Accordingly, in `MergOPT` we sample a perturbation vector from this Laplace distribution at each optimization step to simulate merge-induced parameter offsets. To further assess the sensitivity to the choice of distribution, we additionally experiment with sampling task vectors from a Gaussian distribution, i.e., at each step we draw perturbations from a Gaussian instead of a Laplace. As shown in Table 22, both choices lead to consistent gains over standard fine-tuning. For example, under TIES-Merging, the average performance of `MergOPT` w/ Gaussian and `MergOPT` w/ Laplace when merging two tasks is 0.4912 and 0.4929, respectively, both higher than the baseline value of 0.4854. This indicates that Gaussian sampling can also serve as a reasonable approximation for task vectors, even though it is slightly weaker than the better-motivated Laplace-based approximation.

### D.4.2 BATCH SIZES

In this part, we analyze the sensitivity of the proposed method to the batch size. In our main experiments, we use a default batch size of 8, and here we additionally evaluate batch sizes of 4 and 16. As shown in Table 23, `MergOPT` improves over standard fine-tuning by 6.60% and 1.53% when the batch size is 4 and 8, respectively. When the batch size is increased to 16, we observe a slight decrease of 1.04%. Overall, across a reasonable range of batch sizes, `MergOPT` remains competitive and often yields clear gains over the standard fine-tuning baseline.

### D.4.3 LEARNING RATES

In this part, we analyze the sensitivity of the proposed method to the choice of learning rate. In the main experiments, we use a default learning rate of $2 \times 10^{-5}$. Here, we additionally consider two

Table 18: Performance Comparison of Model Merging Methods on Llama-3.2-1B-Instruct (SGD as base optimizer).

| Method | Task Performance | | | | | | | Avg. (↑) |
|---|---|---|---|---|---|---|---|---|
| | C-STANCE | FOMC | MeetingBank | ScienceQA | NumGLUE-cm | NumGLUE-ds | 20Minuten | |
| Pre-Trained | 0.3386 | 0.2581 | 0.2036 | 0.6780 | 0.1220 | 0.1646 | 0.3802 | 0.3064 |
| SGD Fine-Tuned | 0.4920 | 0.5423 | 0.3630 | 0.8605 | 0.2439 | 0.3415 | 0.3834 | 0.4609 |
| `MergOPT` Fine-Tuned | 0.4710 | 0.6129 | 0.2969 | 0.7694 | 0.1463 | 0.4207 | 0.3933 | 0.4444 |
| Weight Averaging | 0.4023 | 0.3407 | 0.1700 | 0.6268 | 0.3659 | 0.2622 | 0.3762 | 0.3634 |
| **Weight Averaging w/ `MergOPT`** | 0.3912 | 0.3185 | 0.1535 | 0.6351 | 0.2439 | 0.3049 | 0.3804 | 0.3468(-4.56%) |
| Task Arithmetic | 0.4068 | 0.2782 | 0.1382 | 0.5983 | 0.2439 | 0.2561 | 0.3788 | 0.3286 |
| **Task Arithmetic w/ `MergOPT`** | 0.3722 | 0.4556 | 0.1279 | 0.5888 | 0.2439 | 0.3293 | 0.3783 | 0.3566(+8.52%) |
| TIES-Merging | 0.4169 | 0.2782 | 0.1368 | 0.5914 | 0.2195 | 0.2378 | 0.3767 | 0.3225 |
| **TIES-Merging w/ `MergOPT`** | 0.3712 | 0.4476 | 0.1286 | 0.5791 | 0.2439 | 0.3354 | 0.3808 | 0.3552(+10.14%) |
| DARE | 0.3393 | 0.2449 | 0.0938 | 0.3073 | 0.0976 | 0.1585 | 0.3779 | 0.2313 |
| **DARE w/ `MergOPT`** | 0.3164 | 0.2883 | 0.0906 | 0.4131 | 0.0732 | 0.1768 | 0.3710 | 0.2471(+6.83%) |

Table 19: Performance Comparison of Merged Models Trained with Different Fine-Tuning Methods on Llama-3.2-1B-Instruct.

| Method | NumGLUE-cm | NumGLUE-ds | Avg. (↑) |
|---|---|---|---|
| Task Arithmetic (AdamW+AdamW) | 0.3659 | 0.4329 | 0.3994 |
| Task Arithmetic (AdamW+MergOPT) | 0.3659 | 0.4573 | 0.4116(+3.05%) |
| **Task Arithmetic (MergOPT+MergOPT)** | 0.3659 | 0.4817 | 0.4238(+6.10%) |
| TIES-Merging (AdamW+AdamW) | 0.3415 | 0.4390 | 0.3903 |
| TIES-Merging (AdamW+MergOPT) | 0.3659 | 0.4450 | 0.4055(+3.89%) |
| **TIES-Merging (MergOPT+MergOPT)** | 0.3659 | 0.4817 | 0.4238(+8.58%) |

variants by halving and doubling the learning rate to $1 \times 10^{-5}$ and $4 \times 10^{-5}$, respectively. As shown in Table 24, `MergOPT` consistently outperforms standard fine-tuning under all three learning-rate settings. For example, when the learning rate is set to $1 \times 10^{-5}$, $2 \times 10^{-5}$, and $4 \times 10^{-5}$, `MergOPT` yields relative improvements of 2.92%, 1.53%, and 0.32%, respectively, over the Task Arithmetic baseline. These results suggest that our method is robust to reasonable variations in the learning rate.

Table 20: Performance of different merging methods on 2-task groups (Llama-3.2-1B-Instruct).

| Method | Group 1 | | | Group 2 | | |
|---|---|---|---|---|---|---|
| | C-STANCE | FOMC | Avg. (↑) | NumGLUE-cm | NumGLUE-ds | Avg. (↑) |
| Task Arithmetic | 0.4475 | 0.5181 | 0.4828 | 0.3659 | 0.4329 | 0.3994 |
| **Task Arithmetic w/ `MergOPT`** | 0.4440 | 0.5363 | 0.4901(+1.51%) | 0.3659 | 0.4817 | 0.4238(+6.12%) |
| TIES-Merging | 0.4445 | 0.5262 | 0.4854 | 0.3415 | 0.4390 | 0.3902 |
| **TIES-Merging w/ `MergOPT`** | 0.4414 | 0.5444 | 0.4929(+1.54%) | 0.3659 | 0.4817 | 0.4238(+8.62%) |

Table 21: Performance of different merging methods on 6-task groups (Llama-3.2-1B-Instruct).

| Method | Group 1 | | | | | | |
|---|---|---|---|---|---|---|---|
| | C-STANCE | FOMC | MeetingBank | ScienceQA | NumGLUE-cm | NumGLUE-ds | Avg. (↑) |
| Task Arithmetic | 0.4284 | 0.5101 | 0.2136 | 0.7380 | 0.2439 | 0.3537 | 0.4146 |
| **Task Arithmetic w/ `MergOPT`** | 0.4237 | 0.5101 | 0.2115 | 0.7785 | 0.3415 | 0.3537 | 0.4365(+5.28%) |
| TIES-Merging | 0.4288 | 0.4899 | 0.2090 | 0.7400 | 0.2195 | 0.3598 | 0.4078 |
| **TIES-Merging w/ `MergOPT`** | 0.4199 | 0.5121 | 0.2099 | 0.7695 | 0.3415 | 0.3598 | 0.4354(+6.78%) |

| Method | Group 2 | | | | | | |
|---|---|---|---|---|---|---|---|
| | C-STANCE | FOMC | MeetingBank | NumGLUE-cm | NumGLUE-ds | 20Minuten | Avg. (↑) |
| Task Arithmetic | 0.4242 | 0.5363 | 0.2177 | 0.2683 | 0.3476 | 0.3811 | 0.3625 |
| **Task Arithmetic w/ `MergOPT`** | 0.4221 | 0.5363 | 0.2139 | 0.3171 | 0.3537 | 0.3877 | 0.3718(+2.57%) |
| TIES-Merging | 0.4242 | 0.5181 | 0.2171 | 0.2439 | 0.3476 | 0.3789 | 0.3550 |
| **TIES-Merging w/ `MergOPT`** | 0.4215 | 0.5302 | 0.2143 | 0.3171 | 0.3537 | 0.3873 | 0.3707(+4.42%) |

Table 22: Performance Comparison of Different Task Vector Sampling Methods for 2-Task Groups on Llama-3.2-1B-Instruct.

| Method | C-STANCE | FOMC | Avg. (↑) |
|---|---|---|---|
| Task Arithmetic | 0.4475 | 0.5181 | 0.4828 |
| **Task Arithmetic (`MergOPT` w/ Gaussian)** | 0.4440 | 0.5363 | 0.4901(+1.51%) |
| **Task Arithmetic (`MergOPT` w/ Laplace)** | 0.4485 | 0.5320 | 0.4903(+1.55%) |
| TIES-Merging | 0.4445 | 0.5262 | 0.4854 |
| **TIES-Merging (`MergOPT` w/ Gaussian)** | 0.4452 | 0.5372 | 0.4912(+1.19%) |
| **TIES-Merging (`MergOPT` w/ Laplace)** | 0.4414 | 0.5444 | 0.4929(+1.54%) |

Table 23: The Effect of Different Batch Sizes on MergOPT Performance on Llama-3.2-1B-Instruct.

| Method | Batch Size=4 | | | Batch Size=8 (Default) | | | Batch Size=16 | | |
|---|---|---|---|---|---|---|---|---|---|
| | C-STANCE | FOMC | Avg. (↑) | C-STANCE | FOMC | Avg. (↑) | C-STANCE | FOMC | Avg. (↑) |
| Task Arithmetic | 0.4570 | 0.4879 | 0.4725 | 0.4475 | 0.5181 | 0.4828 | 0.4530 | 0.5222 | 0.4876 |
| **Task Arithmetic w/ `MergOPT`** | 0.4610 | 0.5464 | 0.5037(+6.60%) | 0.4440 | 0.5363 | 0.4902(+1.53%) | 0.4570 | 0.5081 | 0.4825(-1.04%) |

Table 24: The Effect of Different Learning Rates on MergOPT Performance on Llama-3.2-1B-Instruct.

| Method | lr = 1e-5 | | | lr = 2e-5 (Default) | | | lr = 4e-5 | | |
|---|---|---|---|---|---|---|---|---|---|
| | C-STANCE | FOMC | Avg. (↑) | C-STANCE | FOMC | Avg. (↑) | C-STANCE | FOMC | Avg. (↑) |
| Task Arithmetic | 0.4270 | 0.5020 | 0.4645 | 0.4475 | 0.5181 | 0.4828 | 0.4805 | 0.5600 | 0.5203 |
| **Task Arithmetic w/ `MergOPT`** | 0.4260 | 0.5302 | 0.4781(+2.92%) | 0.4440 | 0.5363 | 0.4902(+1.53%) | 0.4790 | 0.5650 | 0.5220(+0.32%) |

