# OpenReview forum: "MergOPT: A Merge-Aware Optimizer for Robust Model Merging"
_ICLR.cc/2026/Conference — ICLR 2026 Poster_

### Official Review · Reviewer_emci · 2025-11-01

**Soundness:** 3
**Presentation:** 2
**Contribution:** 3
**Rating:** 6
**Confidence:** 4

**Summary:**

This paper proposes MergOPT, a merge-aware optimization method that enhances the compatibility and robustness of fine-tuned models for subsequent model merging. The method reformulates fine-tuning as a distributionally robust optimization problem in the weight space, where parameters from other expert models are regarded as adversarial perturbations. By introducing controlled perturbations during the fine-tuning stage, MergOPT guides models toward flatter and more stable loss landscapes, thereby improving their adaptability under potential merging operations. In particular, the article designs a fine-tuning optimization method that enables the fine-tuned model to remain more compatible with potential future model-merging operations, thereby achieving better performance when merging occurs. Overall, this work aims to establish a more merge-friendly fine-tuning ecosystem for future model merging, providing a feasible approach to improve merge performance through enhanced finetuned model compatibility.

**Strengths:**

1. The paper introduces a new perspective by shifting the focus of model merging from a post-merging adjustment problem to a fine-tuning optimization problem. This ''merge-aware fine-tunin`` view is conceptually novel and provides a promising new direction for improving model compatibility before merging.

2. The article aims to build a more friendly model ecosystem for future parameter-level merging, allowing users to obtain more compatible expert models during the fine-tuning stage. It is essentially an infrastructure method for constructing a merge-friendly ecosystem, establishing the foundation for future collaborative fine-tuning and model merging workflows.

3. The paper includes extensive analysis on perturbation parameters (e.g., Laplace scale, number of tasks, and merge coefficient) and provides insights into how these factors influence the merge robustness and final model quality.

4. The study demonstrates strong generality, showing that merge-aware fine-tuning can benefit multiple architectures and merging algorithms."

**Weaknesses:**

1. The method assumes that robustness to future, unseen tasks can be achieved by simulating “potential merging perturbations” using only data from single-task fine-tuning. This assumption is overly strong because it depends on the empirical task-vector distribution estimated from the current benchmark and may not generalize to unseen tasks. The diversity of real-world merging scenarios can far exceed what a single parameter distribution can approximate. Moreover, the Laplace assumption mainly captures sparsity or concentration in parameter differences but fails to represent semantic conflicts or gradient-direction contradictions between tasks. Consequently, the assumed perturbation distribution may deviate substantially in cases of stronger conflicting or different modal tasks.

2. Although Eq. (6) defines a robustness objective against worst-case perturbations, the implementation (Eq. 10) adopts the single-step approximation. This procedure perturbs only within a local noise neighborhood instead of solving the true inner maximization. Hence, the method effectively behaves as a heuristic regularization rather than a strict distributionally robust optimization. No theoretical justification or upper/lower bounds are provided to show that the sampled perturbation covers the genuine worst-case scenario. The robustness may thus fail when the future merging tasks deviate from the training-time task-vector distribution.

3. Since MergOPT requires access to fine-tuning data, its applicability in privacy-restricted or cross-organizational settings remains unclear. For example, if user A fine-tunes a model using MergOPT but user B fine-tunes without it, how would the merging behave? Does the method require all participants to adopt MergOPT to realize its benefits?

4. The paper cites recent works such as AdaMerging but does not experimentally compare with them. Including comparisons with newer or optimization-driven merging strategies such as WUDI Merging or RegMean would make the evaluation more convincing.

5. All experiments were conducted on text-based tasks. Extending the evaluation to other modalities, such as vision, would better demonstrate the generality and impact of the proposed method across different modalities.

6. It is recommended that the “Related Work” and “Introduction” sections include more discussion on current fine-tuning methods. For example, an analysis of why previous fine-tuning methods result in poor compatibility during model merging would be valuable. Furthermore, it would be beneficial to review any existing work that is similar to the proposed method and to detail their approaches. This would better align with the positioning of this research.

7. Although SAM results are reported, the discussion could be more prominently integrated into the main text rather than deferred to the appendix, as SAM represents the most closely related optimization paradigm to MergOPT."

**Questions:**

Q1. The Laplace distribution used for perturbation sampling is fitted from a specific set of task vectors. How should this distribution be re-estimated or adapted for new domains, modalities, or highly conflicting task collections? Is there any online or adaptive estimation strategy?
Q2. In real-world collaboration, contributors may fine-tune their models with different strategies. Must all participants adopt MergOPT to benefit from improved mergeability, or can MergOPT-trained and standard fine-tuned models be merged effectively?
Q3. Beyond the parameters explicitly discussed in the paper, does MergOPT exhibit sensitivity to factors such as batch size, learning-rate schedule, or training steps, similar to the SAM?
Q4. What specific experimental device or environment were the results reported in the paper based on? This is crucial for promoting reproducibility.
Q5. What are the significant differences in the experimental settings between Tables 8-9 and Tables 10-11 in the appendix? Why is the latter considered a fairer comparison? Did Tables 10-11 adjust the hyperparameters of SAM to align with those of MergOPT in order to match the computational costs?
Q6. As shown in Tables 8-11 in the appendix, MergOPT and the SAM method actually have their own advantages and disadvantages in terms of performance when combined with different merging methods and in terms of runtime under different configurations.What are the significant differences between these two optimization methods in terms of optimization objectives and perturbation modeling? What inspirations did MergOPT draw from SAM?"

**Details Of Ethics Concerns:**

No ethical issues identified.

---

> ### Author Response · Authors · 2025-11-20
>
> Thank you for your valuable suggestions. Following your advice, we have revised the manuscript and highlighted the changes in blue. Below, we respond point by point to each of the concerns and issues you raised.
>
> ## W1: On the assumption about the distribution of task vectors
> ***Revision note: Please see lines 969–971 on page 18 of the manuscript.***
>
> Indeed, a simple Laplace distribution cannot perfectly approximate the distribution of task vectors. Therefore, the goal of MergOPT is not to construct a distribution that can precisely generate all task vectors, but rather to introduce an approximate perturbation. As long as this distribution shares key properties with the true task-vector distribution, such as zero mean, heavy tails, and a sharp peak, it can help the model become more robust to weight perturbations induced by the merging operation.
>
> ## W2: On the approximate single step perturbation
> ***Revision note: Please see Section 3.2.3, lines 264 to 267 on page 5 and lines 307 to 309 on page 6 of the manuscript.***
>
> In MergOPT, the inner optimization problem simultaneously involves the task vectors $\{\Delta \theta_i\}_{i=1}^K$, the merging coefficients $\alpha$, and the number of merged tasks $K$. During single task fine-tuning, all these variables are essentially unknown to the current expert model. Under this setting, we cannot construct the true inner optimization objective, nor can we deterministically maximize it. Even in an ideal scenario where these variables were fully observable, the combinatorial space of their possible configurations would still be extremely large. To identify the true worst case solution, one would in principle need to enumerate all parameter combinations and fully evaluate each of them on the validation set, which is computationally prohibitive in practice. Therefore, our goal is not to obtain the exact theoretical minimax solution, but to build a structured, sampleable robust approximation that can cover the most critical directions.
>
> Based on our empirical analysis of task vector distributions, we find that task vectors can be reasonably approximated by a Laplace distribution. We therefore sample a perturbation vector $z$ from this distribution and combine it with a pre constructed discrete feasible set over the task number $K$ and merging coefficients $\alpha$, which are jointly sampled at each parameter update. As training proceeds, the region covered by the accumulated samples gradually expands, and in a statistical sense increasingly approximates the true yet unknown distribution of task vectors.
>
> ## W3 \& Q2: On whether models fine-tuned by different methods can be merged
> ***Revision note: Please see Table 17 on page 23 and the description in Appendix D.2.3.***
>
> We sincerely appreciate this important question. We agree that in practical cross‐organization or privacy‐restricted settings, it is often unrealistic to assume that all participants would fine-tune models using MergOPT. Motivated by this observation, we added a new controlled experiment on Llama-3.2-1B-Instruct. In this setting, user A fine tunes the model using MergOPT, while user B applies standard AdamW fine-tuning. We then merge the two resulting models and compare the outcome with (1) both sides using standard fine-tuning (AdamW + AdamW) and (2) both sides using MergOPT (MergOPT + MergOPT). As shown in the table, merging a model fine-tuned with AdamW and one fine-tuned with MergOPT yields better performance than merging two standard AdamW models, though it remains slightly below the performance achieved when both models are fine-tuned using MergOPT.
>
> | Method | NumGLUE-cm | NumGLUE-ds | Avg. |
> |--------|------------|------------|------|
> | Task Arithmetic (AdamW+AdamW) | 0.3659 | 0.4329 | 0.3994 |
> | Task Arithmetic (AdamW+MergOPT) | 0.3659 | 0.4573 | 0.4116 |
> | Task Arithmetic (MergOPT+MergOPT) | 0.3659 | 0.4817 | 0.4238 |
> | TIES-Merging (AdamW+AdamW) | 0.3415 | 0.4390 | 0.3903 |
> | TIES-Merging (AdamW+MergOPT) | 0.3659 | 0.4450 | 0.4055 |
> | TIES-Merging (MergOPT+MergOPT) | 0.3659 | 0.4817 | 0.4238 |

---

> ### Author Response · Authors · 2025-11-20
>
> ## W4: On comparing with the latest baselines
> ***Revision note: Please see Table 7 on page 18 and Section C.3 on page 17 of the manuscript.***
>
> Following your suggestion, we compared our method with the more recent WUDI-Merging [1]. As shown in the table, when merging ten tasks using WUDI-Merging, the proposed MergOPT fine tuning strategy achieves a 1.32% relative improvement over standard fine-tuning. **This demonstrates that our method remains effective even under the latest model merging strategy.**
>
> | Method | SUN397 | Cars | RESISC45 | EuroSAT | SVHN | GTSRB | MNIST | DTD | Flowers102 | PCAM | Avg. |
> |--------|--------|------|----------|---------|------|-------|-------|-----|------------|------|------|
> | WUDI-Merging | 67.5 | 72.6 | 84.0 | 93.8 | 89.4 | 95.8 | 99.2 | 61.2 | 77.5 | 91.2 | 83.2 |
> | **WUDI-Merging w/ MergOPT** | 70.2 | 69.9 | 86.6 | 92.9 | 90.7 | 97.3 | 99.0 | 63.8 | 79.4 | 93.0 | **84.3**(+1.32%) |
>
> [1] "Whoever started the interference should end it: Guiding data-free model merging via task vectors." ICML (2025).
>
>
> ## W5: On evaluation for vision tasks
> ***Revision note: Please see Table 7 on page 18 and Section C.3 on page 17 of the manuscript.***
>
> Following your suggestion, we added experiments on the ViT-B/32 architecture to validate the effectiveness of our method when merging models trained on ten visual tasks. As shown in the table, when our MergOPT is applied to the five merging strategies Weight Averaging, Task Arithmetic, TIES-Merging, DARE and WUDI-Merging, it yields performance gains of 2.36%, 3.43%, 4.10%, 3.58% and 1.32% respectively. **These results confirm that our method remains effective for vision architectures and tasks**.
>
> | Method | SUN397 | Cars | RESISC45 | EuroSAT | SVHN | GTSRB | MNIST | DTD | Flowers102 | PCAM | Avg. |
> |--------|--------|------|----------|---------|------|-------|-------|-----|------------|------|------|
> | Weight Averaging | 60.7 | 54.4 | 59.0 | 35.9 | 33.4 | 33.2 | 58.7 | 42.0 | 77.5 | 94.3 | 54.9 |
> | **Weight Averaging w/ MergOPT** | 60.8 | 56.6 | 59.7 | 36.9 | 32.9 | 35.5 | 66.9 | 42.6 | 77.5 | 92.3 | **56.2**(+2.36%) |
> | Task Arithmetic | 62.1 | 57.0 | 72.0 | 77.7 | 64.5 | 59.0 | 91.4 | 46.3 | 68.6 | 70.5 | 66.9 |
> | **Task Arithmetic w/ MergOPT** | 57.0 | 55.5 | 70.0 | 71.8 | 76.6 | 74.6 | 95.7 | 47.3 | 63.7 | 80.0 | **69.2**(+3.43%) |
> | TIES-Merging | 64.6 | 64.9 | 70.6 | 74.7 | 62.8 | 55.4 | 92.0 | 44.1 | 65.7 | 62.8 | 65.8 |
> | **TIES-Merging w/ MergOPT** | 56.2 | 58.1 | 70.0 | 66.7 | 78.0 | 73.0 | 96.7 | 45.2 | 64.7 | 75.8 | **68.5**(+4.10%) |
> | DARE | 61.1 | 56.9 | 71.5 | 78.5 | 65.6 | 56.7 | 92.1 | 45.2 | 67.6 | 71.6 | 67.0 |
> | **DARE w/ MergOPT** | 59.3 | 55.4 | 70.2 | 74.2 | 74.8 | 73.2 | 95.1 | 48.4 | 64.7 | 78.4 | **69.4**(+3.58%) |
> | WUDI-Merging | 67.5 | 72.6 | 84.0 | 93.8 | 89.4 | 95.8 | 99.2 | 61.2 | 77.5 | 91.2 | 83.2 |
> | **WUDI-Merging w/ MergOPT** | 70.2 | 69.9 | 86.6 | 92.9 | 90.7 | 97.3 | 99.0 | 63.8 | 79.4 | 93.0 | **84.3**(+1.32%) |
>
>
> ## W6: On the description and comparison of more fine-tuning methods
> ***Revision note: Please see lines 64 to 75 on page 2, lines 130 to 136 on page 3, and Section 4.3 on page 9 of the manuscript.***
>
> In the introduction and related work, we have added more discussion and comparison of different fine-tuning methods. We have also added Section 4.3, which provides a detailed comparison with prior SAM based fine-tuning approaches [1,2] and tangent space fine-tuning [3]. As shown by the results, **our method consistently outperforms existing fine-tuning methods in terms of performance.**
>
> | Method | Avg. |
> |--------|----------|
> | Weight Averaging w/ SAFT-Merge |0.3923 |
> | **Weight Averaging w/ MergOPT**| 0.4083 (+4.08%) |
> | Task Arithmetic w/ SAFT-Merge  | 0.4136 |
> | **Task Arithmetic w/ MergOPT**   | 0.4165 (+0.70%)  |
> | TIES-Merging w/ SAFT-Merge|  0.4084 |
> | **TIES-Merging w/ MergOPT**   |0.4123 (+0.95%)  |
> | DARE w/ SAFT-Merge |0.3988 |
> | **DARE w/ MergOPT**   |0.4147 (+3.98%)  |
>
> | Method |  Avg.  |
> |--------|-------- |
> | Weight Averaging (Standard) |  54.9 |
> | Weight Averaging (Tangent) |  50.5(-8.01%) |
> | **Weight Averaging (MergOPT)** |   56.2(+2.36%) |
> | Task Arithmetic (Standard) | 66.9 |
> | Task Arithmetic (Tangent) | 69.0(+3.13%) |
> | **Task Arithmetic (MergOPT)** | 69.2(+3.43%) |
> | TIES-Merging (Standard) | 65.8 |
> | TIES-Merging (Tangent) | 68.4(+3.95%) |
> | **TIES-Merging (MergOPT)** | 68.5(+4.10%) |
> | DARE (Standard) |  67.0 |
> | DARE (Tangent) |  68.7(+2.53%) |
> | **DARE (MergOPT)** |  69.4(+3.58%) |
>
>
> [1] Foret, Pierre, et al. "Sharpness-aware minimization for efficiently improving generalization." ICLR, 2021.
>
> [2] Lee, Yeoreum, Jinwook Jung, and Sungyong Baik. "Mitigating parameter interference in model merging via sharpness-aware fine-tuning." ICLR, 2025.
>
> [3] Ortiz-Jimenez et al.,Task Arithmetic in the Tangent Space: Improved Editing of Pre-Trained Models. NeurIPS 2023.

---

> ### Author Response · Authors · 2025-11-20
>
> ## W7: On placing the SAM results in the main text
> ***Revision note: Please see Section 4.3 on page 9 of the manuscript.***
>
> Thank you for the suggestion. We have added a new Section 4.3 in the main text to provide a detailed comparison between our fine tuning method and SAM based approaches.
>
> ## Q1: Perturbation sampling strategy for new domains and tasks
> ***Revision note: Please see Section D.4.1 on page 23 and Table 20 of the manuscript.***
>
> Task vectors typically follow a zero mean distribution with a sharp peak and heavy tails. For a new task, we recommend directly approximating the task vectors with either a Gaussian or Laplace distribution. To this end, we evaluate the effect of using Gaussian and Laplace distributions within MergOPT. The results are summarized in the table. We observe that, whether using Laplace or Gaussian, MergOPT consistently brings stable performance improvements under two representative merging methods, Task Arithmetic and TIES-Merging. This shows that MergOPT does not strictly require the true task vector distribution to be exactly Laplacian and that similar gains remain when a nearby distribution is used. As expected, the closer the approximation is, as in the case of Laplace compared with Gaussian, the improvement is naturally more pronounced.
>
> | Method | C-STANCE | FOMC | Avg. |
> |---|---|---|---|
> | Task Arithmetic | 0.4475 | 0.5181 | 0.4828 |
> | Task Arithmetic w/ MergOPT (Gaussian) | 0.4440 | 0.5363 | 0.4901(+1.51%) |
> | Task Arithmetic w/ MergOPT(Laplace) | 0.4485 | 0.5320 | 0.4903(+1.55%) |
> | TIES-Merging | 0.4445 | 0.5262 | 0.4854 |
> | TIES-Merging w/ MergOPT (Gaussian) | 0.4452 | 0.5372 | 0.4912(+1.19%) |
> | TIES-Merging w/ MergOPT (Laplace) | 0.4414 | 0.5444 | 0.4929(+1.55%) |
>
>
> ## Q3: About hyperparameters in MergOPT
> ***Revision note: Please see Tables 21 and 22 on page 24 and the descriptions in Appendices D.4.2 and D.4.3.***
>
> In Appendix D.4, we have added an analysis of MergOPT with respect to hyperparameters such as learning rate and batch size. Overall, our method remains effective across different choices of learning rate and batch size.
>
> ## Q4: Experimental setup or environment
> ***Revision note: Please see lines 819–821 on page 16 of the manuscript.***
>
> We have added a description of the main experimental environment in the implementation details of Appendix B.2.
>
> ## Q5: On a fair comparison with SAM
> ***Revision note: Please see Section D.2.1 on pages 20–21 of the manuscript.***
>
> Yes, our MergOPT method uses the same hyperparameters as SAM. For a fair comparison, we further evaluate both methods under two settings: the same number of training epochs (Tables 12–13) and the same optimization time (Tables 14–15). The results show that our MergOPT approach is both more efficient and more effective.
>
>
> ## Q6: Key differences between SAM and MergOPT
> ***Revision note: Please see lines 136 to 142 on page 3 of the manuscript.***
>
> SAM and MergOPT are similar in that both introduce perturbations to the parameters being fine-tuned. However, SAM performs gradient ascent on the current task, which substantially increases the training cost and makes it 2.04 times that of standard fine-tuning. In contrast, MergOPT is better aligned with the model merging setting. Its perturbation directions are sampled from a feasible set jointly defined by the task vectors, the number of tasks, and the merging coefficients, which leads to higher efficiency. The experimental results also show that our method is more effective.

---

### Official Review · Reviewer_VdQC · 2025-11-01

**Soundness:** 3
**Presentation:** 3
**Contribution:** 2
**Rating:** 2
**Confidence:** 4

**Summary:**

The paper tackles model merging, particularly during the fine-tuning stage, similar to few previous works. The paper takes a similar perspective with a previous work that views merging process as parameter perturbation process. From this perspective, the paper proposes a new optimizer, named MergOPT that employs a distributionally robust optimization (DRO). The experiments on three LLM demonstrate the effectiveness of the proposed method, in comparison to using standard fine-tuning in terms of the performance of merged model.

**Strengths:**

- The proposed method is simple yet practical, avoiding the double-forward/backward passes required by SAM and thereby achieving the efficiency.
- The proposed method demonstrates strong performance, validated on three LLM models.
- The paper is well-organized and easy to follow.
- The paper shows that a task vector can be well-approximated by a Laplacian distribution, which provides a new insight.

**Weaknesses:**

- The paper claims that one of core ideas of the proposed method is viewing merging process as perturbations. However, to the best of the reviewer's knowledge, the view of seeing other experts as adversarial perturbation seems to be similar to the view introduced by SAFT-Merge, a work mentioned in the paper. The mathematical formulation in Eq. (5) seems to be similar to the one in SAFT-Merge as well. However, the paper does not provide credit to the previous work for the perspective and formulation.
- The introduction section lacks discussions on previous works that also focus on fine-tuning stage for merging, including but not limited to SAFT-Merge and [A,B,C].
- The related work section lacks discussions on similar previous works [A,B,C].
- The experimental results lack comprehensive comparisons against previous works [A,B,C].
- The experiments are mainly performed only on seven language tasks, in contrast to previous model merging works that also show experiments on vision tasks to demonstrate the applicability and generalizability .

[A] Tang et al., Parameter Efficient Multi-task Model Fusion with Partial Linearization. ICLR 2024.
[B] Ortiz-Jimenez et al.,Task Arithmetic in the Tangent Space: Improved Editing of Pre-Trained Models. NeurIPS 2023.
[C] Jin et al., Fine-Tuning Attention Modules Only: Enhancing Weight Disentanglement in Task Arithmetic. ICLR 2025.

**Questions:**

- The datasets used in the experiment seem to be different from standard datasets used in the field of model merging. What's the justification for such setting? How does the performance compare against previous works on fine-tuning methods on standard datasets?

---

> ### Author Response · Authors · 2025-11-20
>
> Thank you for your valuable suggestions. We have revised the manuscript accordingly and highlighted the changes in blue. Below, we respond to each of the problems you raised one by one.
>
> ## W1: Similarity to the perturbation formulation in SAFT-Merge
> ***Revision Note: See Equation (5) and lines 206–209 on page 4.***
>
> Yes, the idea of treating other experts as adversarial perturbations is related to SAFT-Merge [D]. In the original manuscript, we already discussed SAFT-Merge as the most relevant work in the Related Work section, and we also compared it with our method in terms of both performance and computational cost (Tables 8–11 in the original version; Tables 12–15 in the revised version).
>
> Following your suggestion, we now explicitly mention this similarity in the description of Equation (5). Nevertheless, our work and SAFT-Merge differ fundamentally in both interpretation and implementation of parameter perturbations. SAFT-Merge is motivated by the SAM optimizer [E] and aims to improve loss-landscape flatness. In contrast, our approach is inspired by Distributionally Robust Optimization and focuses on enhancing parameter robustness for model merging. Moreover, SAFT-Merge computes the perturbation direction using the gradient of the current task, whereas our perturbation is constructed via sampled task vectors, merging coefficients, and the number of merged models, making it more aligned with the merging scenario.
>
> Finally, under the same number of training epochs, SAFT-Merge incurs approximately 2.04$\times$ the training cost of standard fine-tuning, whereas our method requires only 1.17$\times$, making it significantly more efficient. When training time is comparable, our method also consistently outperforms SAFT-Merge (see our response to Weakness 3).
>
> ## W2: Discussing other fine-tuning-stage methods in the introduction
> ***Revision Note: See lines 64–75 in the revised manuscript.***
>
> Thank you for recommending these related works. In the previous version, we cited SAFT-Merge [D] and tangent-space fine-tuning methods [B, C], but overlooked [A]. Following your suggestion, we have now incorporated a discussion of these works in the related work section. The corresponding revisions and discussion in the introduction are as follows: "To the best of our knowledge, only a few model merging works explicitly focus on the fine-tuning stage. For example, tangent-space fine-tuning methods linearize the model and perform optimization in its tangent space to enhance weight disentanglement[A,B,C], thereby alleviating conflicts during merging. However, inference with such linearized models typically incurs a $2-3\times$ higher computational cost compared to standard models[A]. In another line of work, SAFT-Merge[D] is inspired by sharpness-aware minimization (SAM)[E] and aims to improve mergeability by encouraging flatter loss landscapes during fine-tuning. Yet SAM-based fine-tuning usually doubles the training time relative to standard fine-tuning. Given that most existing approaches focus primarily on the merging stage and that the few methods targeting fine-tuning often come with substantial training or inference overhead, we argue that there is a strong need for a fine-tuning scheme that is both efficient and effective, while further improving the overall performance of model merging."
>
> ## W3: Discussing other fine-tuning–stage methods in Related Work
> ***Revision Note: See the Related Work section on page 3.***
>
> In the previous version, we discussed SAFT-Merge [D] in detail. In the revised manuscript, we further expand the discussion to include [A, B, C]. The corresponding updates in the Related Work section are as follows: "Ortiz-Jimenez et al. [B] first pointed out that weight disentanglement is a key factor for the effectiveness of task-arithmetic-based model merging. Their method amplifies weight disentanglement by linearizing the model and performing fine-tuning in its tangent space [A,B,C]. Nevertheless, inference with the linearized model typically takes about two to three times longer than with its original nonlinear counterpart [A]. In addition, our experimental results show that such approaches are still inferior in performance to the method proposed in this paper. SAFT-Merge [D] enhances mergeability during fine-tuning by applying sharpness-aware minimization. However, its training cost is nearly twice that of standard fine-tuning, making it inefficient for large models or datasets. In contrast, our MergOPT matches the efficiency of a standard optimizer while explicitly simulating merging via cross-expert merge-offsets, thereby improving stability and overall merging performance. It is worth mentioning that while most existing methods have been evaluated on vision models and image classification tasks, our work is conducted in the context of LLMs and text generation tasks."

---

> ### Author Response · Authors · 2025-11-20
>
> ## W4: Comparison with other fine-tuning works in experiments
> ***Revision note: Please see Section 4.3 on page 9, Section C.3 on page 17 and Table 7, as well as Tables 12, 13, 14 and 15 in Section D.2.1 of the manuscript.***
>
> In the previous version of the manuscript, we provided a detailed comparison with the SAFT-Merge [D] method, corresponding to Tables 8, 9, 10 and 11 in the original submission and Tables 12, 13, 14 and 15 in the revised version. The results of merging seven tasks on LLaMA-1B are summarized in the tables. Under the same number of training epochs, SAFT-Merge and MergOPT each have their own strengths, but the training cost of SAFT-Merge is 2.04 times that of standard fine-tuning, whereas MergOPT requires only 1.17 times the cost. Moreover, given the same training budget, MergOPT consistently outperforms SAFT-Merge.
>
> | Method |  Same Training Epochs  | Same Training Cost |
> |--------|----------|----------|
> | Weight Averaging w/ SAFT-Merge | 0.3980 |0.3923 |
> | **Weight Averaging w/ MergOPT** |  0.4083 (+2.59%) | 0.4083 (+4.08%) |
> | Task Arithmetic w/ SAFT-Merge |  0.4260 | 0.4136 |
> | **Task Arithmetic w/ MergOPT** |  0.4165 (-2.24%)  | 0.4165 (+0.70%)  |
> | TIES-Merging w/ SAFT-Merge |  0.4177 |  0.4084 |
> | **TIES-Merging w/ MergOPT** |  0.4123 (-1.29%)  |0.4123 (+0.95%)  |
> | DARE w/ SAFT-Merge |  0.3942 |0.3988 |
> | **DARE w/ MergOPT** | 0.4147 (+5.20%)  |0.4147 (+3.98%)  |
>
>
>
> Given that the work of Ortiz-Jimenez et al. [B] and others converts nonlinear models into linear ones and performs fine-tuning in the tangent space, which is orthogonal to our optimization-level approach, we did not include a direct comparison in the previous version. Following your suggestion, we have now added a comparison with [B]. As shown in the table below, when merging ten tasks on a ViT-B/32 model, MergOPT consistently outperforms the tangent-space fine-tuning method.
>
> | Method |  Avg.  |
> |--------|-------- |
> | Weight Averaging (Standard) |  54.9 |
> | Weight Averaging (Tangent) |  50.5(-8.01%) |
> | **Weight Averaging (MergOPT)** |   56.2(+2.36%) |
> | Task Arithmetic (Standard) | 66.9 |
> | Task Arithmetic (Tangent) | 69.0(+3.13%) |
> | **Task Arithmetic (MergOPT)** | 69.2(+3.43%) |
> | TIES-Merging (Standard) | 65.8 |
> | TIES-Merging (Tangent) | 68.4(+3.95%) |
> | **TIES-Merging (MergOPT)** | 68.5(+4.10%) |
> | DARE (Standard) |  67.0 |
> | DARE (Tangent) |  68.7(+2.53%) |
> | **DARE (MergOPT)** |  69.4(+3.58%) |
>
>
> ## W5: On vision tasks
> ***Revision note: Please see Section C.3 on page 17 and Table 7 on page 18 of the manuscript.***
>
> Following your suggestion, we have added model merging experiments on standard visual classification benchmarks. Specifically, we merge models trained on the following ten tasks using ViT-B/32: SUN397, Cars, RESISC45, EuroSAT, SVHN, GTSRB, MNIST, DTD, Flowers102 and PCAM. As shown in the table below, our method delivers consistent performance improvements across different model merging schemes.
>
> | Method | SUN397 | Cars | RESISC45 | EuroSAT | SVHN | GTSRB | MNIST | DTD | Flowers102 | PCAM | Avg.  |
> |--------|--------|------|----------|---------|------|-------|-------|-----|------------|------|----------|
> | Weight Averaging  | 60.7 | 54.4 | 59.0 | 35.9 | 33.4 | 33.2 | 58.7 | 42.0 | 77.5 | 94.3 | 54.9 |
> | **Weight Averaging w/ MergOPT** | 60.8 | 56.6 | 59.7 | 36.9 | 32.9 | 35.5 | 66.9 | 42.6 | 77.5 | 92.3 | 56.2(+2.36%)  |
> | Task Arithmetic  | 62.1 | 57.0 | 72.0 | 77.7 | 64.5 | 59.0 | 91.4 | 46.3 | 68.6 | 70.5 | 66.9 |
> | **Task Arithmetic w/ MergOPT** | 57.0 | 55.5 | 70.0 | 71.8 | 76.6 | 74.6 | 95.7 | 47.3 | 63.7 | 80.0 | 69.2(+3.43%) |
> | TIES-Merging | 64.6 | 64.9 | 70.6 | 74.7 | 62.8 | 55.4 | 92.0 | 44.1 | 65.7 | 62.8 | 65.8 |
> | **TIES-Merging w/ MergOPT** | 56.2 | 58.1 | 70.0 | 66.7 | 78.0 | 73.0 | 96.7 | 45.2 | 64.7 | 75.8 | 68.5(+4.10%) |
> | DARE  | 61.1 | 56.9 | 71.5 | 78.5 | 65.6 | 56.7 | 92.1 | 45.2 | 67.6 | 71.6 | 67.0 |
> | **DARE w/ MergOPT** | 59.3 | 55.4 | 70.2 | 74.2 | 74.8 | 73.2 | 95.1 | 48.4 | 64.7 | 78.4 | 69.4(+3.58%)  |
>
>
> ## Q1: Regarding the choice of datasets
>
> Our selection criteria were based on prevailing text-generation tasks while ensuring sufficient diversity in task types. Therefore, the datasets we used cover: (i) Domain-specific applications, such as ScienceQA, FOMC, and MeetingBank. (ii) Multilingual understanding tasks, such as C-STANCE and 20Minuten. (iii) Mathematical reasoning tasks, such as NumGLUE and NumGLUE-cm.
>
> In addition, following your suggestion, we also validated our method on commonly used image datasets in the model-merging literature. Our approach achieves consistently stable improvements, as detailed in our response to W5.

---

> ### Author Response · Authors · 2025-11-20
>
> References
>
> [A] Tang et al., Parameter Efficient Multi-task Model Fusion with Partial Linearization. ICLR 2024.
>
> [B] Ortiz-Jimenez et al.,Task Arithmetic in the Tangent Space: Improved Editing of Pre-Trained Models. NeurIPS 2023.
>
> [C] Jin et al., Fine-Tuning Attention Modules Only: Enhancing Weight Disentanglement in Task Arithmetic. ICLR 2025.
>
> [D] Lee, Yeoreum, Jinwook Jung, and Sungyong Baik. "Mitigating parameter interference in model merging via sharpness-aware fine-tuning." ICLR, 2025.
>
> [E] Foret, Pierre, et al. "Sharpness-aware minimization for efficiently improving generalization." ICLR, 2021.

---

> ### Author Response · Authors · 2025-11-25
> **Looking Forward to Your Feedback on Submission 23513**
>
> Dear Reviewer VdQC,
>
> Thank you very much for the many valuable suggestions you provided during the review process, especially regarding the related work. Although our previous manuscript had already cited most of the works you mentioned, following your advice we have further expanded the discussion and comparisons, which helped us better position our contribution and improve the overall quality of the paper. In the revised version, we have added more detailed discussions and comparisons of methods used in the fine-tuning stage in the Introduction, Related Work, and Experiments sections. We have also extended our evaluation to include more visual tasks and advanced baselines. For each of your comments, the corresponding revisions in the manuscript have been clearly indicated in our response.
>
> Given that the AC encouraged reviewer–author communication before November 25, we would like to sincerely ask whether the current revision has addressed your concerns.
>
> Sincerely,
>
> Authors of Submission 23513

---

> > ### Comment · Reviewer_VdQC · 2025-11-27
> >
> > I appreciate the authors' rebuttal that has addressed some of the concerns. But, the remaining concern is the lack of extensive experimental comparisons against similar fine-tuning works ([A,B,C] and SAFT-Merge). The extensive comparisons against related works could greatly help understand the contributions and effectiveness of the proposed method. And such comparisons could have been included in the main paper (not appendix) to provide better context to the readers. Also, when reading SAFT-Merge and the Appendix of this paper, SAFT-Merge uses ASAM while the authors have compared against SAFT-Merge based on SAM. Why is there a difference?

---

> > > ### Author Response · Authors · 2025-11-29
> > > **Regarding the placement of experimental results and implementation details**
> > >
> > > ***Revision note: Entire Section 4.3 (Lines 421–473 on pp. 8–9) and Tables 14, 15, 16, and 17 in Appendix D.2.1. The modified parts are highlighted in purple.***
> > >
> > > Dear Reviewer VdQC,
> > >
> > > Thank you for this helpful suggestion. In the original submission, we placed the comparison with fine-tuning methods in the appendix mainly due to **page limitations**, which prevented us from including these results in the main text.
> > >
> > > In the revised version, we have added a new Section 4.3 (Lines 421–473 on pages 8–9). Specifically, we moved all comparison experiments with fine-tuning methods (i.e., tangent-space fine-tuning methods [A, B, C] and the SAM-based SAFT-Merge) into the main text as Tables 4 and 5, along with the corresponding discussion. The algorithm pseudocode and other auxiliary details have been moved to the appendix (page 16).
> > >
> > > In addition, regarding your question on whether SAFT-Merge [D] should use SAM [E] or ASAM [F]: SAFT-Merge builds on the idea of flatness-aware optimization to improve the flatness of the loss landscape, which is beneficial for model merging. SAM is the most classical implementation of this idea. To better address your concern, in the revised version we further include the experimental results of SAFT-Merge (ASAM). Concretely, we compare SAFT-Merge (SAM), SAFT-Merge (ASAM), and MergOPT from two perspectives: (a) all methods are trained for the same number of epochs; (b) all methods are trained under a comparable training-time budget, where the SAM-based methods use roughly half the number of epochs.
> > >
> > > We observe that under the same number of epochs (Table (a)), SAFT-Merge (ASAM) is generally superior to SAFT-Merge (SAM), and MergOPT and SAFT-Merge (SAM) each have their own advantages under different merging schemes. However, in this setting the training costs of SAFT-Merge (ASAM) and SAFT-Merge (SAM) are 2.06× and 2.04× that of standard AdamW fine-tuning, respectively, whereas MergOPT incurs only 1.17×, which makes MergOPT more efficient. Under the comparable-time setting in Table (b), MergOPT consistently outperforms SAFT-Merge (SAM), thus being more effective under the same training-time budget, while SAFT-Merge (SAM) and SAFT-Merge (ASAM) remain competitive under different merging strategies.
> > >
> > > In summary, **compared with SAFT-Merge (SAM) and SAFT-Merge (ASAM), our MergOPT can achieve comparable performance with a significantly lower training cost, or achieve better performance under a comparable cost**.
> > >
> > >
> > >
> > > Table (a): setting with the same number of training epochs
> > > | Method                     | Weight Averaging        | Task Arithmetic        | TIES-Merging            | DARE                    | Time   |
> > > |----------------------------|-------------------------|------------------------|-------------------------|-------------------------|--------|
> > > | (a) SAFT-Merge (SAM)       | 0.3980                  | 0.4260                 | 0.4177                  | 0.3942                  | 2.04×  |
> > > | (a) SAFT-Merge (ASAM)      | 0.4047 (+1.68%)         | 0.4119 (−3.30%)        | 0.4226 (+1.17%)         | 0.4101 (+4.03%)         | 2.06×  |
> > > | **(a) MergOPT (Ours)**        | 0.4083 (+2.59%)     | 0.4165 (−2.24%)    | 0.4123 (−1.29%)     | 0.4147 (+5.20%)     | 1.17×  |
> > >
> > > Table (b): setting with comparable training time
> > > | Method                     | Weight Averaging        | Task Arithmetic        | TIES-Merging            | DARE                    | Time   |
> > > |----------------------------|-------------------------|------------------------|-------------------------|-------------------------|--------|
> > > | (b) SAFT-Merge (SAM)       | 0.3923                  | 0.4136                 | 0.4084                  | 0.3988                  | 1.04×  |
> > > | (b) SAFT-Merge (ASAM)      | 0.3868 (−1.40%)         | 0.4181 (+1.08%)        | 0.4030 (−1.32%)         | 0.3943 (−1.12%)         | 1.04×  |
> > > | **(b) MergOPT (Ours)**        | 0.4083 (+4.08%)     | 0.4165 (+0.70%)   | 0.4123 (+0.95%)     | 0.4147 (+3.98%)     | 1.17×  |
> > >
> > > References:
> > >
> > > [A] Tang et al., Parameter Efficient Multi-task Model Fusion with Partial Linearization. ICLR 2024.
> > >
> > > [B] Ortiz-Jimenez et al.,Task Arithmetic in the Tangent Space: Improved Editing of Pre-Trained Models. NeurIPS 2023.
> > >
> > > [C] Jin et al., Fine-Tuning Attention Modules Only: Enhancing Weight Disentanglement in Task Arithmetic. ICLR 2025.
> > >
> > > [D] Lee, Yeoreum, Jinwook Jung, and Sungyong Baik. "Mitigating parameter interference in model merging via sharpness-aware fine-tuning." ICLR, 2025.
> > >
> > > [E] Foret, Pierre, et al. "Sharpness-aware minimization for efficiently improving generalization." ICLR, 2021.
> > >
> > > [F] Kwon, Jungmin, et al. "Asam: Adaptive sharpness-aware minimization for scale-invariant learning of deep neural networks." ICML. PMLR, 2021.
> > >
> > > Sincerely,
> > >
> > > Authors of Submission 23513

---

### Official Review · Reviewer_wjmi · 2025-11-01

**Soundness:** 3
**Presentation:** 3
**Contribution:** 3
**Rating:** 6
**Confidence:** 3

**Summary:**

The paper proposes MergOPT, a fine-tuning-time optimizer that models downstream parameter-level merging as weight-space perturbations and trains each expert to be robust to those perturbations. Experiments on three LLM bases (Llama-3.2-1B/3B, Qwen-2.5-1.5B) and four merging schemes (WA, Task Arithmetic, TIES, DARE) show consistent average gains when merging seven experts, visualizations suggest flatter joint-loss landscapes.

**Strengths:**

1. Shifts attention from *merging-time* heuristics to *fine-tuning-time* preparation, the weight-space DRO view is intuitive and connects to robustness.
2. The single-step perturbation keeps cost close to standard fine-tuning and is optimizer-agnostic.
3. Tables 1–3 show improvements for WA/TA/TIES/DARE; 3.5% average, up to 9.5% in a seven-expert setting.
4. Heatmaps indicate flatter, wider low-loss regions after MergOPT, supporting the robustness claim.

**Weaknesses:**

1. While Figure 1 and Figure 3 empirically show "reasonable",  Laplace fits for task vectors, the choice is ultimately justified mostly visually and via mean/scale parameter tuning. There is no mathematical analysis or statistical test quantifying goodness-of-fit (e.g., K–S distance, log-likelihood comparison to alternatives), nor discussion of the method’s sensitivity to non-Laplacian deviations, skew, or multi-modal effects. As this modeling choice affects the entire robust optimization, more rigor is needed.
2. The practical DRO surrogate in Algorithm 1 replaces the inner maximization over merging perturbations with a simple Monte Carlo sampling from the approximated feasible region. There is little analysis of the gap between this heuristic and true minimax-optimality, particularly for non-Laplacian/real-world merging scenarios. How often does the sampled perturbation land in truly adversarial directions (i.e., the “tails”)? Is there a risk of systematically missing impactful merging failures?
3. The choice of the Laplace distribution, the settings $\mu=0$, $b$ values, and the discrete set $\mathcal{A}$ for $\alpha$ are justified by empirical distributions and prior work, but ablation results (see Table 6, Table 7) could be improved with more diagnostic measures: e.g., what happens if a different parametric family is used (Gaussian, mixture, etc.)? How robust are improvements if empirical task vector statistics deviate (e.g., outlier tasks, long tails, multi-modal structure)?

**Questions:**

Refer to the weakness

---

> ### Author Response · Authors · 2025-11-20
>
> Thank you for your valuable suggestions. We have revised the manuscript accordingly and highlighted the changes in blue. Below, we provide point-by-point responses to each of the issues you raised.
>
> ## W1: Rationality of using the Laplace distribution
> ***Revision Note: See Table 8 on page 19, Table 10 on page 20, and Appendix D.1.3.***
>
> In the revised manuscript, we provide more systematic statistical analyses to strengthen the justification of this modeling choice. Following the reviewer’s suggestion, we randomly sampled three tasks and evaluated the fit of the Laplace distribution using the Kolmogorov–Smirnov (K–S) distance and average log-likelihood. The K–S distances fall within the range of 0.10–0.11, indicating reasonably good overall consistency between the empirical and Laplace distributions. The log-likelihood values also show a stable trend, suggesting that the Laplace distribution provides a satisfactory approximation of the main shape of task vector distributions.
>
> It is important to emphasize that **MergOPT does not use the Laplace distribution to precisely model the full true distribution of task vectors**. Instead, it serves as a simple, efficiently sampled approximation to construct a worst-case merge-offset, enabling distributionally robust optimization in weight space.
>
>
> | Task | K-S Distance | Avg. Log Likelihood |
> |------|--------------|---------------------|
> | C-STANCE | 0.101 | 5.76 |
> | MeetingBank | 0.106 | 5.87 |
> | ScienceQA | 0.106 | 6.54 |
>
>
>
> To further assess the sensitivity of MergOPT to the hyperparameters of the Laplace distribution $\text{Laplace}(\mu, b)$, we evaluate the robustness of our method under different choices of $\mu$ and $b$. Specifically: (i) in Table 9 of the manuscript, we study how varying b affects performance when merging 7 tasks. As shown in the table, the results are similar across different values of $b$, with the best performance obtained at b = 0.0005.
> | $b$ |  Avg.  |
> |----------|----------|
> |0.0005 (default)|0.4164|
> |0.01|0.4119|
> |0.05|0.4137|
>
> (ii)  In addition, in the revised version (Table 10), we further examine the stability with respect to $\mu$ when merging 2 tasks using Task Arithmetic and the TIES-Merging methods. Taking TIES-Merging as an example, the results show that performance is again similar across different values of $\mu$, with $\mu = 0$ performing best.
> | $\mu$ |  Avg.  |
> |----------|----------|
> |-0.1|0.4910|
> |0.0 (default)|0.4929 |
> |+0.1|0.4912|
>
> These results indicate that **our method is relatively robust to the choices of $\mu$ and $b$**.
>
>
> ## W2: About sampled merge-induced perturbations
> ***Revision Note: See Section 3.2.3; lines 264–267 on page 5; lines 307–309 on page 6.***
>
> Since the inner optimization simultaneously involves the task vectors $\{\Delta \theta_i\}_{i=1}^K$, the merging coefficient $\alpha$, and the number of merged models $K$, all of these variables are inherently unknown to the current expert during single-task fine-tuning. Under this setting, we cannot construct the true inner objective, nor can we deterministically maximize it. Even if these variables were fully accessible in an ideal case, the combinatorial search space would still be prohibitively large: obtaining the exact worst-case solution would require exhaustively testing all parameter combinations on a validation set, which is computationally impractical in real-world deployment. Therefore, our goal is not to solve the theoretical minimax objective exactly. Instead, we aim to construct a structured, sampleable, and directionally informative robust approximation.
>
> Based on our empirical analysis of task-vector distributions, task vectors can be reasonably approximated by a Laplace distribution. Thus, we sample merge-offset vectors $z$ from this distribution. In parallel, we sample $\alpha$ and $K$ from pre-defined discrete feasible domains during each parameter update. As training proceeds, the accumulated samples gradually expand the coverage of potential perturbation directions, thereby approximating the true—but unknown—task-vector distribution in a statistical sense.

---

> ### Author Response · Authors · 2025-11-20
>
> ## W3: On sampling task vectors from different distributions
> ***Revision Note: See Table 20 on page 24 and the description in Appendix D.4.1.***
>
> Thank you for the suggestion. In the revised manuscript, we include additional experiments comparing different sampling distributions for task vectors, specifically Gaussian and Laplace, within MergOPT. The results are summarized as follows: regardless of whether we use a Laplace or Gaussian distribution, MergOPT consistently yields stable performance gains under two representative merging methods, Task Arithmetic and TIES-Merging. This indicates that MergOPT does not critically rely on the task vectors being perfectly Laplacian; even when using a nearby distribution such as Gaussian, the benefits persist. Naturally, the closer the sampling distribution is to the true underlying distribution, the more pronounced the improvement tends to be.
>
> | Method | C-STANCE | FOMC | Avg. |
> |---|---|---|---|
> | Task Arithmetic | 0.4475 | 0.5181 | 0.4828 |
> | Task Arithmetic w/MergOPT (Gaussian) | 0.4440 | 0.5363 | 0.4901(+1.51%) |
> | Task Arithmetic w/MergOPT(Laplace) | 0.4485 | 0.5320 | 0.4903(+1.55%) |
> | TIES-Merging | 0.4445 | 0.5262 | 0.4854 |
> | TIES-Merging w/MergOPT (Gaussian) | 0.4452 | 0.5372 | 0.4912(+1.19%) |
> | TIES-Merging w/MergOPT (Laplace) | 0.4414 | 0.5444 | 0.4929(+1.55%) |

---

### Official Review · Reviewer_5xzh · 2025-11-01

**Soundness:** 3
**Presentation:** 3
**Contribution:** 3
**Rating:** 4
**Confidence:** 4

**Summary:**

The paper proposes MergOPT, a merge-aware optimizer designed to improve model robustness to parameter perturbations during model merging. The key idea is to introduce merge-aware regularization during fine-tuning, formulated under a distributionally robust optimization (DRO) framework. By injecting Laplace-distributed weight perturbations during training, the model becomes more resilient to merging-induced parameter shifts. Experiments on multiple large language models (Llama3, Qwen2.5, etc.) and merging baselines (Weight Averaging, Task Arithmetic, TIES, DARE) show consistent performance gains — typically around +3.5% average and up to +9.5% on certain benchmarks.

**Strengths:**

1. Novel training perspective: The work is among the first to explicitly incorporate merge robustness into the optimization objective, shifting the focus from “how to merge” to “how to train for merging.”
2. Theoretical motivation: The DRO-based formulation provides a principled justification for the merge-aware objective.
3. Empirical breadth: The experiments cover several LLM architectures and multiple merging algorithms, showing consistent performance improvements.

**Weaknesses:**

1. Insufficient Baseline Comparisons: The paper does not compare against optimization-driven merging methods, such as [1] and later variants using evolutionary or gradient-based recipe search. This omission weakens the claim of superiority.
2. Scalability and Efficiency Concerns: While MergOPT is shown to be more efficient than SAM, it still adds a non-trivial 17% computational overhead to the fine-tuning process (Table 9). This cost is incurred for every single expert model that is fine-tuned. The paper does not adequately discuss the scalability of this approach in a real-world scenario where dozens of expert models might need to be trained before merging.
3. Scalability is not verified: Experiments focus on 1B–3B models; results for larger (≥7B) or multimodal models are missing, leaving questions about scalability.

[1] Akiba T, Shing M, Tang Y, et al. Evolutionary optimization of model merging recipes[J]. Nature Machine Intelligence, 2025, 7(2): 195-204.

**Questions:**

1. Have the authors tested combining MergOPT with evolutionary or gradient-based merging recipe optimizers?
2. When merging more than 10 models, does the Laplace noise assumption still approximate the merge perturbation distribution well?

---

> ### Author Response · Authors · 2025-11-20
>
> Thank you for your valuable suggestions. We have revised the manuscript accordingly and highlighted the changes in blue. Below, we provide point-by-point responses to each of the weaknesses and concerns you raised.
>
>
> ## W1 \& Q1: About Baseline Comparisons
> ***Revision Note: Please refer to Table 7 on page 18 and the description in Appendix C.3 (pages 16–17).***
>
> Optimization-driven merging methods are indeed an important direction in model merging research. However, this direction is orthogonal to the focus of our work: such methods aim to optimize how to merge during the merging stage, whereas our work focuses on improving the compatibility of expert models during fine-tuning, so that they can be better merged by a variety of existing strategies.
>
> To address the reviewer’s concern regarding merging coefficients, we additionally verified a grid-search strategy. Specifically, for each task vector, we search its merging coefficient in the range $[-0.5, 1.5]$ with a step size of 0.1. As shown in Figure 2 on page 9, across various merging coefficient combinations, our MergOPT-trained models consistently yield lower loss than baseline models after merging.
>
> Furthermore, our initial choice of merging baselines followed the principles of efficiency, data-free merging, and representativeness. Therefore, we included four widely-used classic strategies: Weight Averaging, Task Arithmetic, TIES-Merging, and DARE. To further strengthen the comparison, we additionally incorporated WUDI-Merging [1], a stronger modern merging baseline known to significantly outperform the aforementioned methods. As shown in blow Table, WUDI-Merging achieves an average performance of 83.2, clearly surpassing the four classic strategies. On top of this, combining WUDI-Merging with our MergOPT further boosts the performance to 84.3.
>
>
> | Method | Avg ACC. |
> |--------|------|
> | Weight Averaging  | 54.9 |
> | **Weight Averaging w/ MergOPT** | **56.2** |
> | Task Arithmetic |   66.9 |
> | **Task Arithmetic w/ MergOPT** |  **69.2** |
> | TIES-Merging | 65.8 |
> | **TIES-Merging w/ MergOPT** |   **68.5** |
> | DARE |  67.0 |
> | **DARE w/ MergOPT** |  **69.4** |
> | WUDI-Merging  |  83.2 |
> | **WUDI-Merging  w/ MergOPT** |  **84.3** |
>
> In summary, **our approach remains effective when compared against finer-grained coefficient search schemes, the most classical model merging methods, and the most advanced model merging methods alike**.
>
> [1] Cheng, Runxi, et al. "Whoever started the interference should end it: Guiding data-free model merging via task vectors." ICML (2025).

---

> ### Author Response · Authors · 2025-11-20
>
> ## W2 \& Q2: On scalability with respect to the number of models and efficiency
> ***Revision Note: See Table 17 on page 23 and the description in Appendix D.2.3; Table 7 on page 18 and the description in Appendix C.3 on page 17.***
>
> Yes, training each expert model with MergOPT takes 1.17$\times$ the time of standard fine-tuning, although this is already significantly lower than the 2.04$\times$ overhead of SAM-based fine-tuning [1,2]. It is also important to note that not all models need to be trained with MergOPT; merging MergOPT-trained models with models obtained via standard fine-tuning is still feasible. As shown in bellow Table, we observe that even when only one of the two models (AdamW+MergOPT) is fine-tuned with MergOPT, the merged model already achieves noticeably better performance than merging two standard AdamW models (AdamW+AdamW). For example, under Task Arithmetic, AdamW+MergOPT attains an average score of 0.4116, compared to 0.3994 for AdamW+AdamW. When both models are fine-tuned with MergOPT, the performance further improves. These results indicate that **MergOPT is compatible with and beneficial for merging models obtained from heterogeneous fine-tuning strategies**.
>
> | Method | NumGLUE-cm | NumGLUE-ds | Avg.  |
> |--------|------------|------------|----------|
> | Task Arithmetic (AdamW+AdamW) | 0.3659 | 0.4329 | 0.3994 |
> | Task Arithmetic (AdamW+MergOPT) | 0.3659 | 0.4573 | 0.4116(+3.05%) |
> | **Task Arithmetic (MergOPT+MergOPT)** | 0.3659 | 0.4817 | 0.4238(+6.10%) |
> | TIES-Merging (AdamW+AdamW) | 0.3415 | 0.4390 | 0.3903 |
> | TIES-Merging (AdamW+MergOPT) | 0.3659 | 0.4450 | 0.4055(+3.89%) |
> | **TIES-Merging (MergOPT+MergOPT)** | 0.3659 | 0.4817 | 0.4238(+8.58%) |
>
>
> Additionally, regarding the scalability of MergOPT, our previous manuscript already evaluated model merging across various numbers of experts: Tables 1–2 merge 7 tasks, Table 3 merges 4 tasks, Table 18 merges 2 tasks, and Table 19 merges 6 tasks. In this revision, we further include experiments merging 10 tasks. As shown below, when merging 10 experts, MergOPT remains effective and yields substantial improvements over the base merging strategies. Therefore, **across different numbers of expert models, our method consistently delivers gains, demonstrating its scalability with respect to the number of tasks**.
>
>
> | Method | SUN397 | Cars | RESISC45 | EuroSAT | SVHN | GTSRB | MNIST | DTD | Flowers102 | PCAM | Avg. |
> |--------|--------|------|----------|---------|------|-------|-------|-----|------------|------|------|
> | Weight Averaging | 60.7 | 54.4 | 59.0 | 35.9 | 33.4 | 33.2 | 58.7 | 42.0 | 77.5 | 94.3 | 54.9 |
> | **Weight Averaging w/ MergOPT** | 60.8 | 56.6 | 59.7 | 36.9 | 32.9 | 35.5 | 66.9 | 42.6 | 77.5 | 92.3 | **56.2**(+2.36%) |
> | Task Arithmetic | 62.1 | 57.0 | 72.0 | 77.7 | 64.5 | 59.0 | 91.4 | 46.3 | 68.6 | 70.5 | 66.9 |
> | **Task Arithmetic w/ MergOPT** | 57.0 | 55.5 | 70.0 | 71.8 | 76.6 | 74.6 | 95.7 | 47.3 | 63.7 | 80.0 | **69.2**(+3.43%) |
> | TIES-Merging | 64.6 | 64.9 | 70.6 | 74.7 | 62.8 | 55.4 | 92.0 | 44.1 | 65.7 | 62.8 | 65.8 |
> | **TIES-Merging w/ MergOPT** | 56.2 | 58.1 | 70.0 | 66.7 | 78.0 | 73.0 | 96.7 | 45.2 | 64.7 | 75.8 | **68.5**(+4.10%) |
> | DARE | 61.1 | 56.9 | 71.5 | 78.5 | 65.6 | 56.7 | 92.1 | 45.2 | 67.6 | 71.6 | 67.0 |
> | **DARE w/ MergOPT** | 59.3 | 55.4 | 70.2 | 74.2 | 74.8 | 73.2 | 95.1 | 48.4 | 64.7 | 78.4 | **69.4**(+3.58%) |
> | WUDI-Merging | 67.5 | 72.6 | 84.0 | 93.8 | 89.4 | 95.8 | 99.2 | 61.2 | 77.5 | 91.2 | 83.2 |
> | **WUDI-Merging w/ MergOPT** | 70.2 | 69.9 | 86.6 | 92.9 | 90.7 | 97.3 | 99.0 | 63.8 | 79.4 | 93.0 | **84.3**(+1.32%) |
>
> [1] Foret, Pierre, et al. "Sharpness-aware minimization for efficiently improving generalization." ICLR, 2021.
>
> [2] Lee, Yeoreum, Jinwook Jung, and Sungyong Baik. "Mitigating parameter interference in model merging via sharpness-aware fine-tuning." ICLR, 2025.

---

> ### Author Response · Authors · 2025-11-20
>
> ## W3: On scalability with respect to parameter size
>
> ***Revision Note: See Table 6 on page 17 and Appendix C.2 on page 16.***
>
> Thank you for the suggestion. We have added experiments on **Llama-3.1-8B-Instruct**. As shown in the table, compared with merging models fine-tuned using standard training, our method yields improvements of 1.08%, 1.40%, 0.46%, and 0.51% under Weight Averaging, Task Arithmetic, TIES-Merging, and DARE, respectively. **These results demonstrate the applicability of our method on larger-scale architectures**.
>
> | Method | C-STANCE | FOMC | ScienceQA | NumGLUE-cm | Avg. ($\uparrow$) |
> |--------|----------|------|-----------|------------|------|
> | Weight Averaging | 0.4982 | 0.6489 | 0.9293 | 0.7319 | 0.7021 |
> | **Weight Averaging w/ MergOPT** | 0.5071 | 0.6627 | 0.9307 | 0.7384 | **0.7097(+1.08%)** |
> | Task Arithmetic | 0.5268 | 0.6731 | 0.9218 | 0.6833 | 0.7012 |
> | **Task Arithmetic w/ MergOPT** | 0.5318 | 0.6803 | 0.9251 | 0.7069 | **0.7110(+1.40%)** |
> | TIES-Merging | 0.5193 | 0.6792 | 0.9212 | 0.7074 | 0.7068 |
> | **TIES-Merging w/ MergOPT** | 0.5236 | 0.6824 | 0.9237 | 0.7106 | **0.7101(+0.46%)**|
> | DARE | 0.5227 | 0.6696 | 0.9168 | 0.7071 | 0.7041 |
> | **DARE w/ MergOPT** | 0.5259 | 0.6765 | 0.9192 | 0.7092 | **0.7077(+0.51%)** |

---

> ### Comment · Reviewer_5xzh · 2025-11-24
>
> Thank you for the extensive and detailed answers. They have addressed all of my concerns. As a result, I am increasing my score.

---

> > ### Author Response · Authors · 2025-11-25
> > **Thank you very much for supporting our work.**
> >
> > Reviewer 5xzh,
> >
> > Thank you very much for your valuable comments regarding the baselines, model scale, and visual tasks. Your suggestions are very helpful for improving the quality of our work. We are also especially grateful for your final support of our paper.
> >
> > Sincerely,
> >
> > Authors of Submission 23513

---

### Official Review · Reviewer_yTnV · 2025-11-05

**Soundness:** 2
**Presentation:** 3
**Contribution:** 3
**Rating:** 6
**Confidence:** 3

**Summary:**

The paper considers the problem of model merging with LLMs. The key idea to propose a new optimization method is that, usually, despite a good merging procedure, fine-tuning processes don't address parameter conflicts between models in a robust way, which decreases performance severely. To solve this problem, the work introduces a merging-aware optimizer inspired by the recent *distributional robust optimization* (DRO) method (Lin et al. 2022), which sets an objective based on training data coming from a perturbed distribution around a divergence constraint (let's say these distributions are perturbed versions of the empirical data density). The authors adapt this idea for the weight-space, where perturbations are now caused by the addition of new models to the merging objective. Despite computational and privacy issues, the paper proposed a manageable objective and tested it on several LLM-oriented datasets.

**Strengths:**

- The manuscript is in good shape, and I can see the research quality of the work at first sight. The paper is excellently written, extremely clear, does a very good review of the needed background and SOTA methods, and derives the solution in a thorough way. One of the key points I enjoyed more while reading is the clarity and transparency on the challenges and how these are addressed (i.e. basically the approach and discussion introduced in section 3.2.3).
- The DRO techniques are quite interesting, and it makes sense to me that authors develop their solution in this direction. Although I missed a bit of an extra teaser-figure or empirical evidence in pp.4 on how DRO works (for the sake of being a self-contained work or better comprehension), the way it works is well explained and connected to the solution.
- The way the method is proposed, with the perturbation-based approach and later the Laplace-fit and sampling, is very proximal to some probabilistic merging approaches which have been mostly ignored due to they have not yet been applied to LLMs, etc. I don't think this is precisely a big weakness, but I just wanted to mention it. The way objectives are presented and discussed is insightful.

**Weaknesses:**

To me, the paper doesn't have very big critical weaknesses that make it difficult to communicate a scientific contribution. However, what I detect are several big gaps of information, derivations, avoidance of certain clarity on tricky issues and some details that should be corrected:

- I can guess how the last terms in Eq. 5 are obtained, but I would like to have/see the entire derivation of it. Where is the baseline/initial parameter set $\theta_{0}$?
- The paper abuses a bit of the word "perturbation". In some way it confuses the reader, as what we have is different arithmetic combinations of parameters. Someone this naming and recursive mentioning decouples the line-of-explanations from the usual model merging clarity and intuition.
- Not a lot of references to the maximum number of K models to merge or the size. These two are key limitations, and I think they should be present in the discussion and development since the beginning of the manuscript.
- The *alternating optimization strategy* mentioned in pp.5 looks a lot like something we have already considered in the optimization literature with focus on ML/AI (i.e. coordinate ascent/descent, min/max or expectation-maximization). I am sure the authors can find and add some extra details in that direction, so it provides more insights to the reader.
- Already in section 3.2 the combinatorial problem of having a lot of different perturbations as K increases (without even considering the size of models), can be deduced easily. This is somehow mentioned lated in section 3.2.3 in a way that is not very precise for this reviewer. Having a combinatorial problem of this sort is a big problem, despite doing sampling later solves it a bit.
- The comments on the innacessibility of models and tasks in pp. 5 should have been considered from the beginning as a big assumption. What about alignment of logits/labels? what are the main features of the models we are considering?
- First paragraph in pp.6: the empirical distribution of weight perturbations doesn't seem like a good fit to a Laplace distribution, but more of an approximation which underestimates a lot the probability mass on the tails.

**Questions:**

- Results are very confusing without explaining what the performance metrics are, not bold numbers and not having insights on its properties (what is better, being higher or lower). Leaving all this little effort to the reader is not the best idea for communicating precisely the empirical results.
- How many times and how is the Laplace distribution fitted to the empirical changes? is the empirical distribution computed per-model or taking into account all K models together?
- Could the authors add some info on the computational/combinatorial cost of the methodology?

---

> ### Author Response · Authors · 2025-11-20
>
> Thank you for your valuable suggestions. We have revised the manuscript accordingly and highlighted the changes in blue. Below, we respond to each of the weaknesses and concerns you raised, one by one.
>
> ## W1: Regarding the derivation of Equation (5)
> ***Revision Note: See Equation (5) on page 4 of the updated manuscript.***
>
> In Equation (5), the pretrained model is implicitly included in $\Delta \theta_i$ because each task vector is defined as $\Delta \theta_i = \theta_i - \theta_0$, i.e., the parameter difference between the fine-tuned model and the initial pretrained model. In addition, we have revised Equation (5) and added intermediate derivation steps to make the formulation clearer.
>
>
>
> ## W2: On the inappropriate use of the term “perturbation”
> ***Revision Note: Revisions have been made throughout the manuscript.***
>
> We thank the reviewer for pointing out the inappropriate use of the term “perturbation”. To avoid potential confusion for readers, we have replaced “perturbation” throughout the paper with suitable alternatives such as “merge-offset” or “merge-induced parameter shift”, which better align with the context of model merging and parameter composition studied in this work.
>
>
> ## W3: On the support for the number and scale of merged models
> ***Revision Note: See lines 294–295 on page 6 of the updated manuscript.***
>
> In Section 3.2.3, when discussing the number of merged models, we have added additional references [1,2,3,4,5] to support the claim that, in the context of LLMs, most model merging studies involve merging only a small number of models. It is important to emphasize that our goal is not to merge an extremely large number of models, but rather to design a better fine-tuning strategy that makes models easier to merge. Therefore, the discussion on the number of models serves only to help specify the range of the hyperparameter $K$ in Equation (10).
>
> [1] Goddard, Charles, et al. "Arcee’s MergeKit: A Toolkit for Merging Large Language Models." EMNLP: Industry Track. 2024.
>
> [2] Du, Guodong, et al. "Parameter competition balancing for model merging." NeurIPS. 2024.
>
> [3] Yu, Le, et al. "Language models are super mario: Absorbing abilities from homologous models as a free lunch." ICML. 2024.
>
> [4] Akiba, Takuya, et al. "Evolutionary optimization of model merging recipes." Nature Machine Intelligence 7.2 (2025): 195-204.
>
> [5] Wan, Fanqi, et al. "Knowledge Fusion of Large Language Models." ICLR. 2024.
>
>
> ## W4: On the connection between the alternating optimization in page 5 and classical optimization problems
> ***Revision Note: See lines 246–249 on page 5 of the revised manuscript.***
>
> The alternating optimization strategy described on page 5 is indeed formally related to classical optimization methods. Specifically, the inner problem in our formulation can be viewed as a max-optimization problem, analogous to a gradient ascent step, while the outer problem is a min-optimization problem, corresponding to a gradient descent step.
>
> However, in the model merging setting considered in this work, our solution strategy also differs from traditional approaches in several key aspects. First, we do not perform multi-step gradient ascent on the inner variables to explicitly solve for an optimal solution. Instead, starting from a robustness-driven objective, we derive a worst-case approximation and perform a one-step sampling from a pre-constructed feasible set, which substantially reduces the computational cost of the inner optimization while preserving robustness (see Section 3.2.3 for details). Second, our min–max structure is specifically tailored to the model merging problem: the inner uncertainty does not correspond to generic adversarial noise, but is encoded as “merging parameters” (Equation (10)) jointly determined by task vectors, merging coefficients, and the number of merged models.
>
> In this sense, while our method is inspired by the classical alternating optimization framework, it incorporates customized modeling and efficient approximations specifically designed for the model merging scenario.
>
> ## W5: On the combinatorial issue of different perturbations
> ***Revision Note: See lines 307–309 on page 6 of the revised manuscript.***
>
> The merging coefficient $\alpha$, the number of merged models $K$, and the task-vector space $\mathcal{Z}$ together define the feasible set of merge-induced offsets. Performing an explicit worst-case maximization over this space would be computationally infeasible due to its combinatorial complexity. Therefore, instead of explicitly solving this maximization problem, we adopt an approximation strategy that performs a single-step sampling to replace the costly inner optimization. This enables us to achieve effective robustness improvements, even when the combinatorial space is extremely large and the theoretical optimum is impractical to compute.

---

> ### Author Response · Authors · 2025-11-20
>
> ## W6: On the inaccessibility of models/tasks and label alignment
> ***Revision Note: See lines 87–88 on page 2 of the revised manuscript.***
>
> The inaccessibility of other expert models and their task vectors is a core assumption in the model merging setting. This assumption is explicitly stated as “Unknowable Merge-Offset Variables” in Sec. 3.2.3, but we agree that it should be emphasized earlier. In the revised version, we have added a clarification in the introduction to make this assumption more prominent.
>
> In addition, parameter-level merging does not rely on logits-level alignment. Since merging is performed in the weight space, all expert models share the same tokenizer, architecture, and output dimensionality (a standard setup in model merging experiments). As a result, the logits naturally have matching dimensions, and no additional alignment procedure is required.
>
>
> ## W7: On the empirical distribution of weight perturbations
> ***Revision Note: See lines 969–971 on page 18 of the revised manuscript.***
>
> We agree that the empirical distribution in the figure does not perfectly match a Laplace distribution, with noticeable deviations, particularly an underestimation of the tail mass. In real deep learning settings, the optimization process is highly complex, and the resulting parameter update distribution is unlikely to be captured exactly by any simple explicit distribution. Therefore, a perfect match is not a practical objective.
>
> In this work, our use of the Laplace distribution is not intended to precisely recover the true distribution of task vectors, but rather to provide a simple, easy-to-sample approximation with sharp peaks and heavy tails. This enables us to construct challenging merge-offset directions while maintaining a tractable formulation.
>
>
> ## Q1: About the evaluation metrics
> ***Revision Note: See line 351 on page 7 and Table 4 on page 15 of the revised manuscript.***
>
> We have added a detailed description of the evaluation metrics for each task in Table 4 of the Appendix, clarifying that all metrics are “higher-is-better”. We also included a corresponding explanation in Section 4.1 of the main text.
>
>
> ## Q2: Clarification on the fitting cases
>
> In Figure 1 of the main text, the fitted distribution corresponds to the sum of task vectors aggregated over K models. In contrast, Figure 3 in the Appendix fits the distribution of a single task vector.
>
> ## Q3: About the computational cost of the proposed method
> In Table 13 of the Appendix, we compare the time cost of classic AdamW optimization, SAM optimization [1,2], and our proposed MergOPT optimization. As shown in the table, MergOPT requires about 1.17$\times$ the optimization time of AdamW, whereas SAM incurs a 2.04$\times$ time cost.
>
> [1] Foret, Pierre, et al. "Sharpness-aware minimization for efficiently improving generalization." ICLR, 2021.
>
> [2] Lee, Yeoreum, Jinwook Jung, and Sungyong Baik. "Mitigating parameter interference in model merging via sharpness-aware fine-tuning." ICLR, 2025.

---

### Comment · Area_Chair_2NSr · 2025-11-23
**Next Steps Following Authors’ Rebuttal: Review Rebuttal and Participate in Discussion**

Dear Reviewers,

Thank you very much for your thoughtful evaluations of this paper.

Now that the authors have submitted their rebuttal, I kindly ask you to take the following steps (if you have not done so already):

- Read the other reviews as well as the authors’ response.
- Consider whether the rebuttal and additional comments affect your assessment of the paper.
- Engage in interactive discussion with the authors **before November 25**, encouraging a dynamic exchange rather than a one-sided rebuttal.

The current reviews for this paper are mixed. Your contributions at this stage are essential for forming a well-informed final decision. I therefore ask that you reassess your views in light of the authors’ responses and the broader discussion among reviewers.

I am happy to join and support the discussions between you and the authors. Please feel free to share your thoughts and participate actively in the discussion.

Thank you once again for your service to ICLR 2026.

Best regards,

 AC

---

### Meta-Review · Area_Chair_AHUq · 2026-01-09

**Summary:**

The paper proposes a new optimizer for fine tuning, with the goal to improve downstream model merging performance when merging several different finetuned models. The final method considers weight perturbation during fine tuning sampled from a Laplace distribution. Experiments show that the method consistently outperforms standard fine tuning, while giving comparable performance to SAM (while being 2x faster).

Reviewers generally liked the contribution, but raised several concerns. An initial concern was lack of comparison to finetuning with SAM (a method called SAFT-Merging), which has been mostly addressed by the rebuttal. A caveat here is that there are follow up papers to SAM which reduce the cost of SAM to that of regular SGD training, making the claimed 2x speed-up benefit argument less strong. Also, there exists a large number of other weight-perturbation methods than SAM, in particular, variational Bayesian learning approaches with Laplace posterior consider similar weight injection mechanisms as the present paper.

Other concerns of reviewers were about scale of the experiments, and details about implementation and derivation of the method. For instance, there are large gaps between the original "ideal" method and what is finally implemented, but these have mostly been addressed by the rebuttal.

Due the above, this is a borderline decision, but reviewers liked many aspects of the paper and the experiments will be interesting to the community. The paper is recommended for acceptance.

**Reviewer Concerns:**

Essentially all reviewer concerns on derivation of the algorithm, comparison to SAM and larger scale experiments have been adequately addressed by the rebuttal.

**Reviewer Scores:**

Reviewers yTnV, 5xzh, wjmi and emci would have all given score 6 or higher.  Reviewer VdQC gave score 2, but given that their concerns were addressed, they would have increased to 4 or 6.

---

### Decision · Program_Chairs · 2026-01-26

Accept (Poster)